



**Changes in population depth distribution and oxygen stratification explain the current**
**low condition of the Eastern Baltic Sea cod (*Gadus morhua*)**
Michele Casini[1,2], Martin Hansson[3], Alessandro Orio[1], Karin Limburg[1,4]
[1] Swedish University of Agricultural Sciences, Department of Aquatic Resources, Institute of
Marine Research, Lysekil, Sweden
[2] University of Bologna, Department of Biological, Geological and Environmental Sciences,
Bologna, Italy
[3] Swedish Meteorological and Hydrological Institute, Gothenburg, Sweden
[4] State University of New York College of Environmental Science and Forestry, Department
of Environmental and Forest Biology, Syracuse, New York, USA
Corresponding author: Michele Casini; e-mail: michele.casini@slu.se





**Abstract**
During the past twenty years, hypoxic areas have expanded exponentially in the Baltic Sea,
which has become one of the largest marine "dead zones" in the world. At the same time, the
most important commercial fish population of the region, the Eastern Baltic cod, has
experienced a drastic reduction in mean body condition, but the processes relating hypoxia to
condition remain elusive. Here we use extensive long-term monitoring data on cod biology and
distribution as well as on hydrological variations, to investigate the processes that relate
deoxygenation and cod condition during the autumn season. Our results show that the depth
distribution of cod has increased during the past four decades at the same time of the expansion,
and shallowing, of the waters with an oxygen concentration known to be detrimental for cod
performance. This has resulted in a spatial overlap between the cod population and low-
oxygenated waters after the mid-1990s, which relates with the observed decline in cod mean
body condition. Complementary analyses on fish otolith microchemistry also revealed that
since the mid-1990s, cod individuals with low condition were indeed exposed to low-oxygen
waters during their life. This study helps to shed light on the processes that have led to a decline
of the Eastern Baltic cod body condition, which can aid the management of this population
currently in distress. Further studies should focus on understanding why the cod population has
moved to deeper waters in autumn and on analysing the overlap with low-oxygen waters in
other seasons to quantify the potential effects of the variations in physical properties on cod
biology throughout the year.

**Keywords**: hypoxia, fish body condition, direct exposure, depth distribution, cod *Gadus*
*morhua*



## 1. Introduction

The oceans and marine coastal areas are experiencing dramatic deoxygenation worldwide (Breitburg et al., 2018). Declining oxygen can have multiple direct and indirect effects on aquatic organisms and entire ecosystems (Breitburg, 2002; Rabalais et al. 2002; Wu, 2002; Diaz and Rosenberg, 2008; Levin et al., 2009). In particular, studies undertaken both in the wild and within experimental set-ups have revealed large effects of hypoxia on basic metabolism, behavior, ecology, distribution and life-history traits of fish (Pichavant et al., 2001; Eby et al., 2005; Herbert and Steffensen, 2005; Domenici et al., 2007; Stramma et al., 2012).

The Baltic Sea (Fig. 1) is one of the largest brackish areas in the world where the oxygenated, yet scarce and irregular saline water inflows from the adjacent North Sea, combined with a water residence time of about 25–30 years, make the system particularly prone to hypoxia (Carstensen et al., 2014). As a consequence, and in combination with global warming and eutrophication, the Baltic Sea has become one of the largest anthropogenic "dead zones" in the world (Breitburg et al., 2018), with well documented degradation or elimination of benthic communities and disruption of benthic food webs over vast areas (Conley et al., 2009). In particular, since the early 1990s the anoxic and hypoxic areas have increased exponentially in the southern and central Baltic Sea (Carstensen et al., 2014) (Fig. 2).

In this degraded demersal and benthic environment, the body condition (a morphometric index of fish fatness and well-being) of the dominant demersal fish population, the Eastern Baltic cod *Gadus morhua* (hereafter simply referred to as Baltic cod), has declined since the mid-1990s (Casini et al., 2016a). This decline has been related to a decrease in the main pelagic prey abundance in the main distribution area of cod (Eero et al., 2012; Casini et al., 2016a), but also to the increased extent of hypoxic and anoxic areas (Casini et al., 2016a). However, the





underlying mechanisms of the relationship between cod condition and hypoxia are still elusive
(but see Limburg and Casini, 2019). The mechanistic processes linking hypoxia and cod
conditions could be various and not mutually exclusive, including stress due to direct hypoxia
exposure, contraction in the spatial distribution of the population, and change in the
surrounding biota such as reduction of important benthic prey (Casini et al., 2016a). A recent
study pointed out the importance of the decline in the feeding level and energy intake of cod
after the mid-1990s, which was explained by the decline in important benthic prey in the
environment (Neuenfeldt et al., 2019). Lately some investigations have also put forward the
hypothesis that the observed changes in the distribution of demersal fish species, including cod,
were due to the variations in the extent of the hypoxic areas in the Baltic Sea (Orio et al., 2019),
although in-depth analyses were not performed to confirm this hypothesis. The low cod
condition in recent decades has been stressed also by the fishery that has lamented an
increasingly high proportion of catches of lean cod with low economic value. Low condition
has a negative effect on reproductive potential (Mion et al., 2018), mortality (Casini et al.,
2016b) and potentially also movements (Mehner and Kasprzak, 2011) with indirect effects on
prey and therefore food-web structure and ecosystem functioning as shown in other systems
(e.g. Ekau et al., 2010). Therefore, it is very important to understand the ultimate factors leading
to low cod condition and in particular the processes explaining the correlation between cod
condition and deoxygenation of the Baltic Sea water over time.
In this study, we further examine the mechanisms linking deoxygenation to cod condition in
the Baltic Sea. We specifically analyse the temporal changes in the depth distribution of cod,
from long-term monitoring data, in relation to the oxygen levels acknowledged in literature to
affect cod behavior and performance. We support these analyses investigating the relation
between fish exposure to hypoxia and cod condition using otolith microchemistry. Fish otoliths
(ear stones) composed of aragonite accrete continually throughout life and incorporate trace





elements, providing a direct, retrospective measure of an individual fish's environmental and
physiological history.

**2. Materials and methods**
**2.1 Biological data and estimation of cod condition**
Biological data on Eastern Baltic cod individuals were collected during the Baltic International
Trawl Survey, BITS, between 1991 and 2018 (retrieved from the DATRAS database of the
International Council for the Exploration of the Sea, ICES; www.ices.dk) and previous
Swedish and Latvian bottom trawl surveys performed in 1979-1990 in the Baltic Sea (Casini
et al., 2016a). Cod individual body condition (Fulton's K) was estimated as $K = W/L^3 * 100$,
where W is the total weight (g) and L the total length (cm) of the fish. Mean condition was
estimated for ICES Subdivision (SD) 25 (corresponding to the main distribution area for cod
since the early 1990s, Orio et al., 2017) and SDs 26-28 separately. Condition was estimated for
small fish (represented here by the size-class 20-29 cm) and large fish (represented here by the
size-class 40-49 cm). We focused on the cod condition in autumn (i.e. quarter 4), corresponding
to the cod main growth season after spawning in spring-summer (Mion et al., 2020). Moreover,
for autumn long time-series of oxygen levels and extent of hypoxic areas are also available
(Casini et al., 2016a).
**2.2 Estimation of cod depth distribution**
Indices of cod biomass (calculated as catch-per-unit-effort, CPUE, kg/h, herein referred to as
biomass) and depth distribution (i.e. mean depth and interquartile range of predicted depth
distribution) from the BITS and historical bottom trawl surveys in SDs 25-28 from 1979 to
2018 were estimated for large ($\geq$ 30 cm) and small cod (15-30 cm) using a modelling procedure
similar to the one used in Orio et al. (2019). However, in the current study rather than including



environmental variables in the models, quarter was included in interactions with latitude and
longitude, and with depth. To estimate the changes in cod depth distribution in SDs 26-28 that
account for the changes in the spatial distribution of the cod population, the SD-specific depth
distributions were weighted by the annual SD-specific cod CPUEs from the bottom trawl
surveys in quarter 4, estimated from the same model.
**2.3 Depth of hypoxic layers**
Baltic cod has been shown to avoid oxygen concentrations below 1 ml/l (approximately 1.4
mg/l) (Schaber et al., 2012). Therefore, time-series of the depth at which 1 ml/l oxygen
concentration was encountered by SD were obtained from the Swedish Meteorological and
Hydrological Institute (SMHI, www.smhi.se).
Time-series of depth at which 4 ml/l oxygen concentration was encountered by SD were also
obtained from SMHI. This oxygen concentration, on average, has been found to affect the
performance of fish (Vaquer-Sunyer and Duarte, 2008). Specifically for cod, 4 ml/l has been
found as threshold under which an effect on condition and growth starts to be observable
(Chabot and Dutil, 1999). Therefore, we expected that the occurrence of cod in areas and depths
with an oxygen concentration $\leq$ 4 ml/l would lead to an increase in the proportion of cod
individuals with very low condition and a decrease in mean condition in the population.
To relate the depths at which 1 ml/l and 4 ml/l oxygen concentrations were encountered to cod
depth of occurrence and condition in SDs 26-28, the oxygen depths by SD were weighted with
the annual SD-specific cod CPUEs from the bottom trawl surveys estimated from the same
models in quarter 4. In this way, the oxygen circumstances in the SDs where cod was more
abundant were weighted the most.
**2.4 Otolith microchemistry**





Otoliths (N = 154) were selected from Baltic cod collected in the study area in the 1980s-2010s
from BITS and historical bottom trawl surveys in February (Limburg and Casini, 2019). These
were cleaned, transversely sectioned, and analysed by laser ablation inductively coupled
plasma spectrometry. A spot of 100-micron diameter was driven at 5 µm/sec, 10 Hz, to create
a transect from the otolith core to the outer dorsal edge, collecting a suite of elements (see
Limburg and Casini, 2018 for details). For the analysis described here, we took the ratio of
manganese to magnesium along this continuous transect. Manganese, although redox-sensitive
and thus available as dissolved $Mn^{2+}$ and $Mn^{3+}$ at low oxygen concentrations, is also affected
by the fish's growth rate (Limburg et al., 2015; suggested by Thomas et al., 2019).  Dividing
manganese by the corresponding, growth-sensitive magnesium (from the same replicate) to
some extent corrects for the growth effect (Limburg and Casini 2018, 2019). Our metric for
hypoxia exposure is the fraction of an annual growth band wherein this Mn/Mg ratio exceeds
an age-based threshold (Limburg and Casini 2018, 2019). We tested this metric as a function
of cod condition categorized into "high" (condition $\geq 0.9$) and "low" (condition $< 0.9$) groups,
and tested whether this had changed over time (before the year 2000, and from 2000 onward).

**3. Results**
**3.1 Cod condition**
Cod condition increased slightly between the mid-1970s and mid-1990s, but declined abruptly
thereafter. This pattern was similar in SD 25 and SDs 26-28 for both small and large cod (Fig.
3), but after the mid-1990s condition dropped more for large cod. The percentage of large fish
with very low condition ($< 0.8$, see Eero et al., 2012) increased from the end of 1990s in both
SD 25 and SDs 26-28 reaching in recent years 30-40%. The percentage of small fish with low
condition also increased, but lagged temporally behind the large cod, and at 10-20% of





observations was lower than the high incidences of large cod in poor condition (Fig. 3). In general, in SD 25 condition declined slightly more (and the percentage of fish with very low condition increased more) than in SDs 26-28 after the mid-1990s.

**3.2 Cod depth distribution**

Large cod in SD 25 were distributed between 30 and 50 m depth (average of 40 m depth) at the beginning of the time-series, but have been found in deeper waters since the late 1990s (Fig. 4A). In SDs 26-28 large cod were distributed between 35 and 55 m depth (average 45 m) at the beginning of the time-series, while afterwards they moved deeper and since the mid-1990s they became distributed between 50 and 70 m depth (average 60 m) (Fig. 4C). Along with the change in mean depth, large cod in SDs 26-28 have shown a contraction of the range of depth distribution in the past 20 years. Small cod were distributed somewhat shallower than the large fish, but also moved into deeper waters during the time period investigated. In SD 25, these shifted distribution from between 30 and 50 m depth (average 40 m depth) to 45-60 m depth (average 53 m) (Fig. 5A). In SDs 26-28 small cod moved deeper with time as well, from 30-50 m depth (average 40 m) to 50-63 m depth (average 55 m), and experienced a contraction of the range of depth distribution similar to what occurred for the large fish in this area (Fig. 5C).

**3.3 Depth of hypoxic layers**

The depth at which 1 ml/l was encountered remained fairly constant at around 70 m in SD 25, while in SDs 26-28 it became shallower from being deeper than 100 m before the early 1990s to 70-80 m in the past twenty years (Fig. 4A,C and 5A,C). The depth at which 4 ml/l was encountered diminished in SD 25 from 60-65 m at the beginning of the time period to 50-55 m during the past twenty years, while in SDs 26-28 it became shallower from being 70-80 m before the early-1990s to 55-60 m in the past fifteen years (Fig. 4A,C and 5A,C). The oxygen





depths in SDs 26-28, accounting for the SD-specific distribution of the cod, did not differ much
between large and small cod (compare Fig. 4C and Fig. 5C).

**3.4 Depth overlap between cod and hypoxic layers**

In SD 25, large cod depth distribution never overlapped with depth with oxygen ≤ 1 ml/l along
the time period analysed, while in SDs 26-28 there was an overlap in a couple toward the end
of the time-series (Fig. 4A,C). On the other hand, large cod distribution heavily overlapped
with the depth with oxygen ≤ 4 ml/l since the mid-1990s (Fig. 4A,C) and the overlap, although
oscillating, increased in the past twenty years reaching values above 50% in SD 25 and up to
100% in SDs 26-28 (Fig. 4B,D).
Small cod distribution never overlapped with depth with oxygen ≤ 1 ml/l along the time period
analysed, neither in SD 25 nor SDs 26-28 (Fig. 5A,C). On the other hand, small cod distribution
overlapped with the depth with oxygen ≤ 4 ml/l since mid-1990s (Fig. 5A,C) and the overlap,
although oscillating, increased in the past fifteen years reaching values higher than 60% both
in SD 25 and SDs 26-28 (Fig. 5B,D).
There was a strong positive correlation between the percentage of the cod population in waters
≤ 4 ml/l and the percentage of cod individuals with very low condition (for large cod, r = 0.71
and 0.74 in SD 25 and SDs 26-28, respectively; for small cod, r = 0.58 and 0.59 in SD 25 and
SDs 26-28, respectively). There was also a strong negative correlation between the percentage
of the cod population in waters ≤ 4 ml/l and mean cod condition (for large cod, r = -0.77 and -
0.76 in SD 25 and SDs 26-28, respectively; for small cod, r = -0.60 and -0.54 in SD 25 and
SDs 26-28, respectively).

**3.5 Otolith microchemistry**

Fish exposed to hypoxia as measured by otolith chemistry showed different responses as a
function of their condition at time of capture and the time period (pre- or post-2000; Fig. 6).



Prior to 2000, the annual duration of hypoxia exposure was relatively low (35.4%); for the
years 2000 and onward, the percent duration rose to 51.8%. More strikingly, when divided
further into groups by fish condition, pre-2000 fish were not significantly different with respect
to hypoxia exposure regardless of condition. After 2000, fish with condition < 0.9 had been
exposed considerably longer to hypoxia (62.7% ± 3.6) than fish with condition ≥ 0.9 (40.9% ±
5.1; Fig. 6). The effect sizes of interaction of time period and condition were large and highly
significant ($F_{1,746} = 23.287$, $p = 2 \times 10^{-6}$).

**4. Discussion**
In this paper, we analysed the potential mechanisms relating Baltic Sea deoxygenation with
changes in Eastern Baltic cod body condition during the past four decades. To this end, we
investigated the changes in depth distribution of the cod population and the vertical changes in
oxygen gradients based on long-term biological and hydrological monitoring data. Moreover,
we supplemented these analyses with proxies for hypoxia exposure from individual fish otolith
microchemistry.
**4.1 Cod depth of distribution and overlap with hypoxic areas**
Our analyses show an increase in the areas with an oxygen level below cod tolerance (i.e.
oxygen ≤ 1 ml/l; Schaber et al., 2012). Moreover, this oxygen threshold has also shifted with
time towards shallower depths, determining an overall contraction of the potentially suitable
habitat for cod (Casini et al., 2016a). Declines in oxygen concentrations have caused a
contraction of the habitat and the distribution of fish in other systems (Eby and Crowder, 2002;
Stramma et al., 2012; Breitburg et al., 2018) with measurable effects on, for example,
individual growth (e.g. Campbell and Rice, 2014). In the Baltic Sea, however, this change
seems not to have affected the cod depth of distribution in autumn, since the latter has been




always above 70-75 m, a depth only in few years reached by the waters with 1 ml/l. On the
other hand, it could be hypothesized that during the latest decade the cod population was unable
to occupy deeper habitats because of the vertical rise of this oxygen layer. This hypothesis
seems to be supported by the decline in the range of depth distribution (i.e. a squeeze of the
cod habitat occupation) shown by both large and small cod in SDs 26-28 during the past twenty
years. Explaining the temporal changes in the depth distribution of cod is beyond the scope of
this paper, but a potential reason could be that cod seek deeper layers to avoid too warm waters,
which could be detrimental when resources are scarce. In fact, pelagic prey have declined after
the mid-1990s in the southern and central Baltic Sea (Casini et al., 2016a) and therefore cod
might go deeper to optimize metabolism. Small cod, moreover, could seek deeper waters to
escape from the predation of the increased seals and aquatic birds (Orio et al., 2019).
The depth where dissolved oxygen falls to ≤ 4 ml/l ("sub-lethal" level, i.e. level that has been
shown in previous studies to affect cod performance; Chabot and Dutil, 1999; Vaquer-Sunyer
and Duarte, 2008) has shallowed during the past four decades, as a consequence of
deoxygenation. Our analysis revealed that this vertical rise, together with the deepening of the
cod depth distribution, has resulted in that cod has started to dwell more and more in these
hostile low-oxygen waters. This is consistent with observations of hypoxia exposure proxied
by otolith chemistry (Limburg and Casini, 2018 and 2019; this study, see below). The overlap
between cod depth distribution and "sub-lethal" oxygen layers occurred and reinforced only
after the mid-1990s, concomitant with the decline in cod condition, while in earlier years the
cod population was occurring above those layers. Therefore, according to our expectations and
hypothesis, the negative effects of hypoxia on cod condition could only arise after the mid-
1990s. This is also in accordance with our otolith microchemistry analysis (see below) and
previous investigations that suggested that in the earlier years (before the mid-1990s) cod
condition was regulated by other factors, such as pelagic prey biomass and density-dependence





(Casini et al., 2016a, Limburg and Casini, 2019). The progressively higher proportion of the
cod population in "sub-lethal" oxygen layers, as revealed by our study, conforms also to the
increasingly higher proportion of individuals in extremely low condition (< 0.8 Fulton's K),
which include starving fish and fish close to the condition mortality threshold (Eero et al., 2012;
Casini et al., 2016b).
**4.2 Otolith microchemistry**
The complementary analyses performed on fish otolith microchemistry confirmed that since
the mid-1990s, cod individuals with low condition were indeed exposed to low-oxygen waters
during their life. Duration of hypoxia exposure as measured in Baltic cod otoliths has increased
markedly since mid-1990s (Limburg and Casini, 2018) and was found in our study to be
significantly greater in fish in poor condition at time of capture. This is a remarkable finding,
given that condition is measured only once during life (at capture), and the observations of
hypoxia exposure are taken throughout life. This suggests that currently, condition may carry
over from chronic exposure to low oxygen, which weakens fish and produces a cascade of
effects, from reduced metabolic scope leading to lower activity and slower digestion (Claireaux
and Chabot, 2016), to greater susceptibility to disease and parasites (e.g., Sokolova et al.,
2018). In contrast, in fish captured prior to 2000 the overall exposure to hypoxia was lower and
showed no relationship with condition. Thus the otolith microchemistry analysis confirmed the
that, pre-2000, factors other than hypoxia played a greater role in shaping cod condition as
concluded also by Casini et al. (2016a).
Although we have shown here that direct oxygen exposure is likely a key factor shaping cod
condition after the mid-1990s, other factors might contribute to explain the decline in condition
as well (Casini et al., 2016a). The more severe decline in condition in SD 25 compared to SDs
26-28, for example, could be due to the higher density of cod in the southern Baltic Sea during



the past twenty years (Orio et al., 2017) leading to density-dependent effects, and the lower
abundance of sprat, the main pelagic fish prey for cod, in this area (Casini et al., 2014).
Moreover, deoxygenation, by deteriorating the benthic communities, has likely affected
negatively important benthic prey for cod and therefore influenced also indirectly cod condition
and growth (Neuenfeldt et al., 2019).
**5. Conclusions**
We have shown here the potential mechanisms linking deoxygenation to cod condition in the
Baltic Sea. A combination of increased depth of distribution of the cod population and a vertical
rise of the "sub-lethal" oxygen layers has led cod dwelling progressively more in hostile low-
oxygen waters, contributing to explain the reduction in cod condition in the past two decades.
Further analyses should focus on revealing the reasons of the shift of cod distribution to deeper
and less-oxygenated waters. We stress that our depth analyses were focused on the autumn
season, when cod growth is maximised for the accumulation of energy reserves to be utilized
for spawning the following spring-summer (Mion et al., 2020). The changes in cod depth of
distribution are different in other seasons, especially those before and during spawning (Orio
et al., 2019), when cod could have different environment requirements for reproduction.
Therefore, further analyses should be performed to investigate the changes in cod population
depth distribution in relation to oxygen stratification in other seasons to better understand the
biotic and abiotic spatio-temporal dynamics, and their effects on cod performance, over the
entire year.






**Data availability**

Time-series used in this study are available upon request to the corresponding author.

**Author contribution**

MC designed and coordinated the study. MC, MH, AO and KL prepared the raw data. MC
estimated cod condition, MH performed the hydrographic modelling, AO performed the cod
distribution modelling, and KL prepared and analysed the cod otoliths. MC prepared the first
draft of the manuscript and all authors contributed to the final version.

**Competing interests**

The authors declare that they have no conflict of interest.

**Acknowledgements**

We thank all the personnel involved in the long-term fish and hydrological monitoring
programmes and data collection at the SLU's Department of Aquatic Resources (and former
Swedish National Board of Fisheries) and at the Swedish Meteorological and Hydrological
Institute. We also thank the Institute of Food safety, Animal Health and Environment "BIOR",
Latvia, for the historical Latvian data on cod condition and survey catches.

**Financial support**

This study was funded by the Swedish Research Council Formas (grant no. 2018-00775 to
Michele Casini: "Fish interactions in the marine benthic habitat: a knowledge gap in Baltic Sea
fish ecology and multispecies fisheries management") and the US National Science Foundation
(project OCE-1923965 to Karin Limburg: "Shifting the hypoxia paradigm – new directions to
explore the spread and impacts of ocean/Great Lakes deoxygenation").



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





**Figure captions**

Fig. 1. Bathymetric Map of the Baltic Sea divided into ICES Subdivisions (SDs). The study area includes the SDs 25–28 (i.e. the Central Baltic Sea).

Fig. 2. Maps of the Baltic Sea with superimposed the areas with oxygen concentration ≤ 1 ml/l (black, avoided by cod) and ≤ 4 ml/l (grey, sub-lethal level, producing negative effects on cod performance) in 1990 (panel A) and 2018 (panel B). Time-series of the total area ($km^2$) with oxygen concentration ≤ 1 ml/l and ≤ 4 ml/l in the SDs 25-28 (panel C). Data were from the Swedish Meteorological and Hydrological Institute (SMHI, www.smhi.se) (see also Casini et al., 2016a).

Fig. 3. Temporal developments of mean cod condition (± 1 s.d.) in Subdivision 25 and Subdivisions 26-28 for small cod (20-29 cm) and large cod (40–49 cm). Superimposed (grey bars) the temporal developments of the percentage of cod with very low condition (< 0.8) for the respective areas and length classes.

Fig. 4. Time-series of large cod (≥ 30 cm) depth distribution (mean and interquartile range of each predicted depth distribution; see Orio et al., 2019) as well as depths of oxygen concentration 1 ml/l and 4 ml/l, for Subdivision 25 (panel A) and Subdivisions 26-28 (panel C). Panels B and D, time-series of the percent of large cod in waters with oxygen concentration ≤ 4 ml/l, in Subdivision 25 and Subdivisions 26-28.

Fig. 5. Time-series of small cod (15-30 cm) depth distribution (mean and interquartile range of each predicted depth distribution; see Orio et al., 2019) as well as depth of oxygen concentration 1 ml/l and 4 ml/l, for Subdivision 25 (panel A) and Subdivisions 26-28 (panel C). Panels B and D, time-series of the percent of small cod in waters with oxygen concentration ≤ 4 ml/l, in Subdivision 25 and Subdivisions 26-28.



Fig. 6. Differences in otolith chemistry as related to hypoxia and fish condition for pre-2000
and 2000s. Within-year hypoxia exposure duration is proxied by the fraction of each annual
growth band in which the otolith Mn/Mg ratio exceeds age-specific thresholds. These are
categorized by condition factor (high condition is 0.9 or greater) measured at time of capture
(see Limburg and Casini, 2019).



















Figure 1

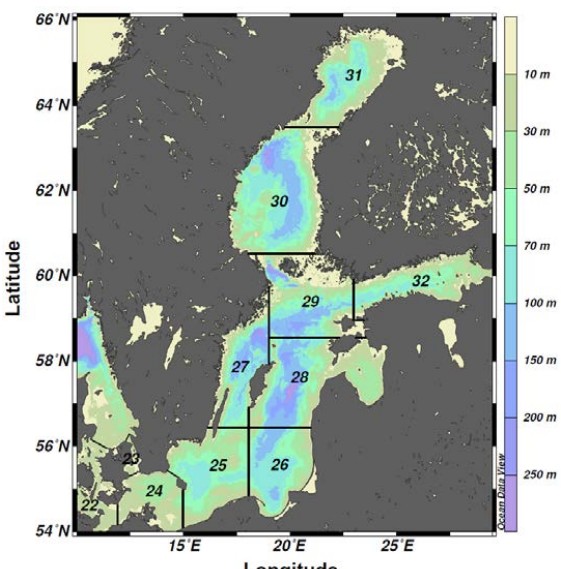
















Figure 2


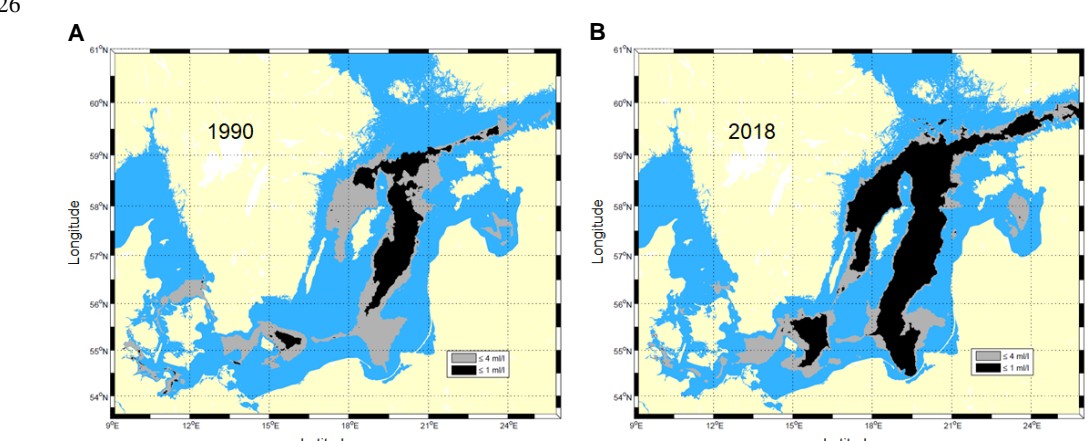











Figure 3

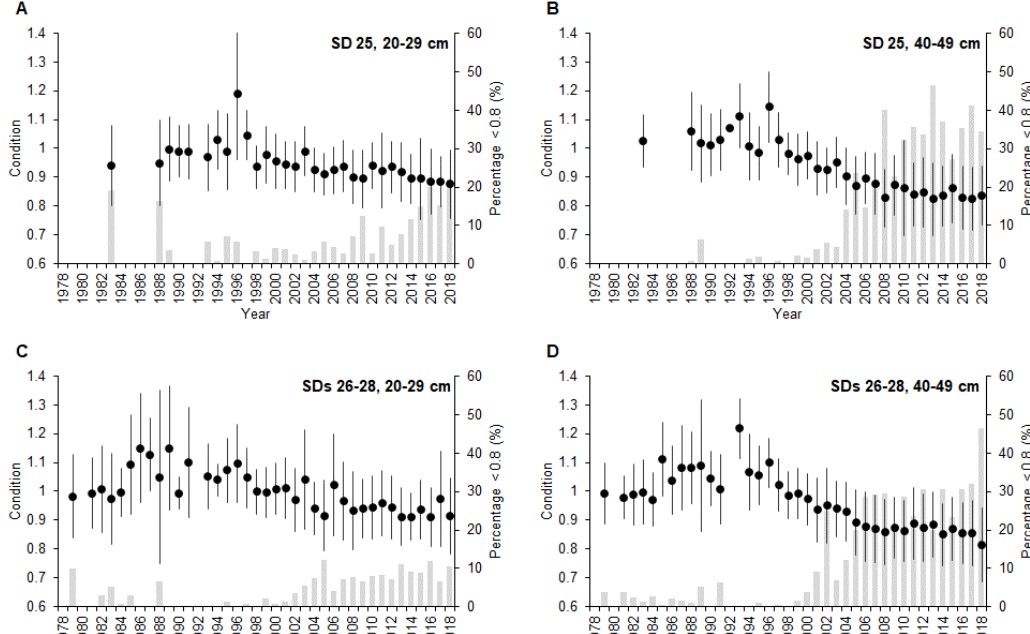















Figure 4

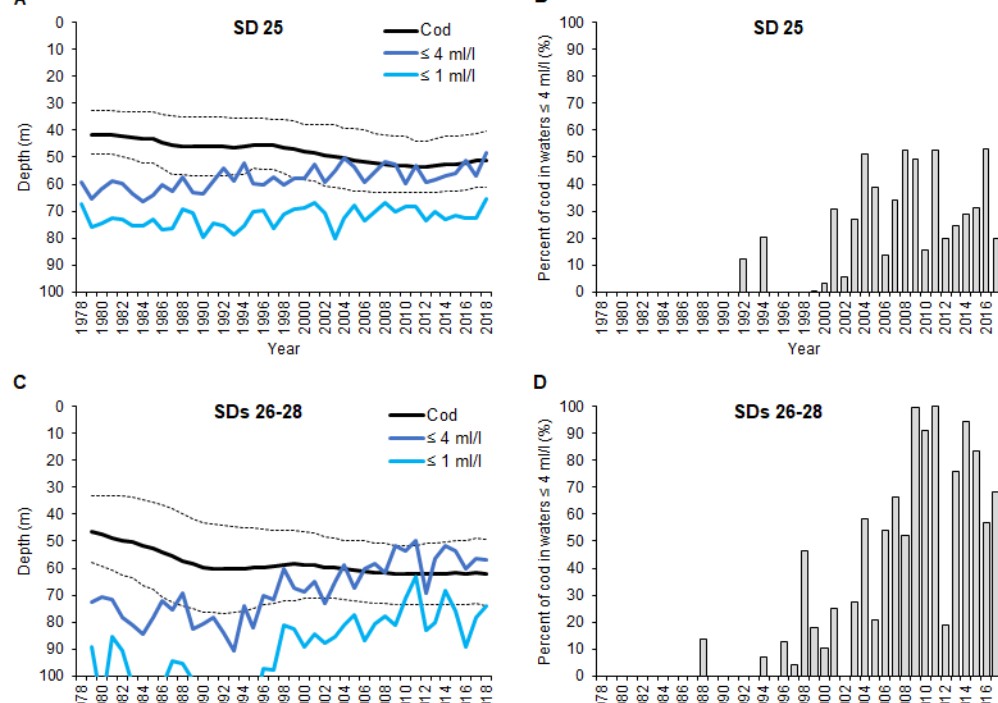














Figure 5

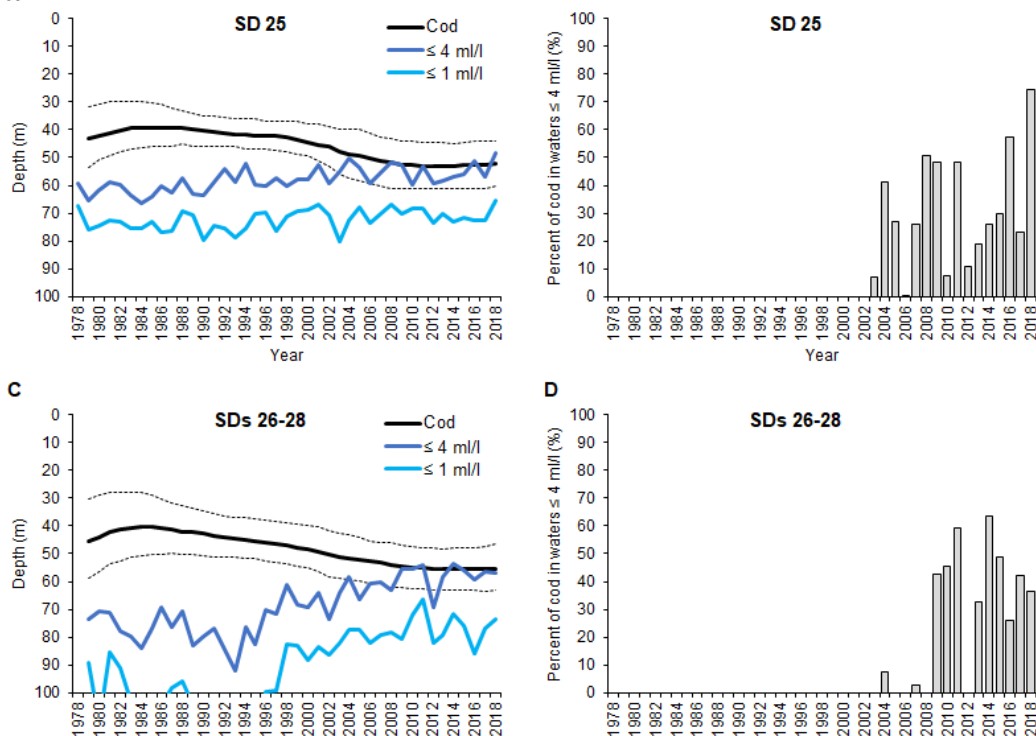





Figure 6

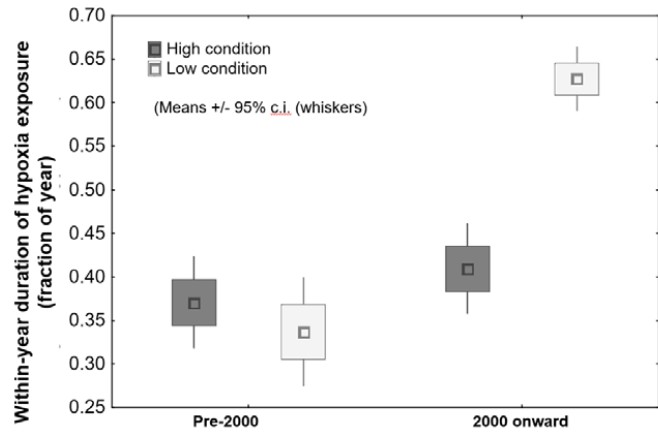




