# Peer review of "Changes in population depth distribution and oxygen stratification are involved in the current"

_Biogeosciences, 2020_

## Referee Comment (RC1) · Keith Brander (Referee) · 1 May 2020

General introduction Decline in oxygen in the oceans due to global change is altering the composition and productivity of marine biota, but the scarcity of long time series, deficient understanding of the processes involved and complex interactions make it difficult to identify the causes of change with confidence. For oxygen, the effects are most evident in enclosed coastal seas, such as the Baltic Sea, where oxygen decline has been observed for several decades due to eutrophication, irregular re-supply of oxygen-rich water from the North Sea and increasing temperature. The Baltic Sea can

in some respects be regarded as a global "canary in the coalmine" for the progressive effects of oxygen decline on fish species and their fisheries. The salinity and oxygen environment is extreme for marine species, such as cod, therefore small changes will quickly have effects and these changes will be detected quickly, because the hydrographic and biological environment is and has been closely monitored for many years. During the past century there have been major fisheries for cod, sprat and herring, but the fishery for cod was closed in 2019 due to the low abundance and poor condition of the fish. In addition to decline in oxygen and salinity and increasing temperature, possible causes for the decline in cod include fishing pressure, parasites, predators and lack of prey as well as combinations or interactions between them. Some of the causes (or pressures) can actually or potentially be altered by management action, but management will only be effective if the processes are correctly identified. This is the frame within which the current paper is important; any lessons learned will be valuable in both the immediate Baltic Sea management context but also as a guide to studying similar cases on a global scale. One of the main lessons however is that even in a low biodiversity system, such as the Baltic Sea, there are many interactions; processes that drive changes during one period of years or decades may be relatively unimportant in another. Subject of the paper - novelty, adequacy and value This paper (I will refer to it by the first letters of the authors names CHOL) relates the observed decline in body condition of cod since the mid-1990s to the observed deepening of their distribution, which is concurrent with observed shallowing of the well-oxygenated upper layer of the water column. Over time the fish are increasingly exposed to oxygen levels that we know from experimental studies to inhibit their growth and condition. A review is required to consider whether the evidence and explanations are novel and whether the analysis and interpretation stand up to peer scrutiny. Much has been published on the causes of the decline in growth and condition of Baltic Sea cod, including a number of prominent recent papers by these authors, so the question of novelty has to be addressed in that context. The decline in condition of cod has been known and commented on for some time, indeed it is one of the reasons that the cod fishery has become uneconomic, as

fillet size and quality decreased. The shallowing of the well-oxygenated upper layer is also well known. The second sentence of the Abstract in CHOL states that "the processes relating hypoxia to condition remain elusive", but the third sentence states that low oxygen levels are "known to be detrimental for cod performance". In fact the underlying physiological processes relating condition and growth to oxygen levels are well known and CHOL base much of their analysis on published experimental work that shows how low oxygen affects cod growth and condition ((Chabot & Dutil, 1999)âĄǎ, henceforth C&D). The novelty of CHOL comes from the way in which they present the relationship between the observed changes in condition and change in vertical distribution of cod related to oxygen levels, which they support with previously published evidence from otolith microchemistry (Limburg & Casini, 2019)âĄǎ. I will go into detail about some of the analysis and interpretation later, since CHOL raise a number of issues that are important in furthering our understanding of what is happening to Baltic Sea cod, however first I will explain why I do not think that the relationship they seek to establish between oxygen and fish body condition adds much to our understanding of the causal relationships and processes involved. It is well know that inferring causal relationships from correlations is slippery for at least two reasons (i) the "chicken and egg" problem, of which an example in the context of Baltic cod is that fish with high parasite loads are in poor condition, but how do we determine whether parasite load causes poor condition or poor condition makes fish more vulnerable to parasites? (ii) the "lurking variable" problem of which a classic example is the correlation between number of daily shark attacks at a beach and number of ice creams sold – both depend on the number of people on the beach. Nevertheless, causal inference is possible and in the case of Baltic Sea cod I suggest that the "lurking" cause of their poor condition has already been identified as reduced food and energy intake over the period since the mid 1990s ((Neuenfeldt et al., 2019)âĄǎ. If there is a "chicken and egg" case for ascribing low feeding to poor condition rather than the reverse, then CHOL should put it forward in their paper. If reduced food and energy intake is indeed a sufficient causal explanation for poor condition then we are of course still left with the question

of what causes reduced feeding, which is an active subject of debate. The principal causes currently put forward include hypoxia (Brander, 2020)⁠ and/or lack of prey for cod (Neuenfeldt 2019), but others include parasites, predators (seals and cormorants) and dietary deficiencies (thiamine), combined in various ways with each other and with density dependence. I would argue that in terms of causal inference, hypoxia has prior status because it is based on known, general processes of metabolism that always apply. Since we know that the condition of cod is affected by their oxygen environment and we have observed an ongoing decline in oxygen in the Baltic Sea, it would be surprising if cod condition did not decline. The issues to resolve are firstly whether cod redistribute themselves to remain in areas and depths with sufficient oxygen and if not then secondly whether the magnitude of ambient oxygen decline that cod experience is sufficient to explain all or only part of the observed change in their condition. If declining oxygen is not a sufficient explanation then other factors, such as availability of suitable food also need to be evaluated. Unfortunately our knowledge of changes in prey, particularly benthos, is poor, compared with our knowledge of changes in oxygen. Evaluating causality and the case made by CHOL CHOL base their argument that declining condition of cod is due to increasing exposure to low oxygen conditions on three sets of evidence: (i) experimental evidence (C&D) that the condition of cod is lower at low oxygen (ii) time series showing concurrent declines in cod condition and their ambient oxygen (which is dependent on changes in depth distributions of both variables) (iii) evidence from otolith microchemistry that cod have been increasingly experiencing low levels of oxygen. As I argued earlier, the experimental evidence is consistent with axiomatic physiological processes. Before evaluating the other two let us consider the use of the term "hypoxia" since it affects the form of analysis and the perception of how oxygen affects cod. "Hypoxic" can apply to a process or to a metric (e.g metabolism, growth, condition) being impaired due to lack of oxygen. In aquatic systems it can also refer to a state of low oxygen (typically <2-3ml/l) that results in mortality of fish and benthos, resulting in "dead" areas or zones. Either continuous impairment or discrete zones, which divide the continuous oxygen spectrum in a water column into two or more

discrete layers, may be used when analysing the effect of low oxygen on fish feeding and condition. CHOL divide the water column into three layers, with a boundary at 4ml of oxygen per l separating the well-oxygenated "normoxic" surface layer from the layer they call "sublethal" or "hostile" and a boundary at 1ml/l dividing this from the "hypoxic" bottom layer, which they state that cod avoid. (In their Discussion they also call everything below the normoxic layer "hypoxic"). This allows them to estimate the change in depth and area of the "hostile" layer and then the change in overlap between this and the layer in which cod occur. I am not convinced that this is the best way to proceed and suggest treating oxygen as a continuous variable with progressive effects on cod, as shown by the experimental results by C&D that they use for defining boundaries. Treating oxygen as three discrete layers may make some forms of analysis easier, but it also raises a number of issues. Firstly, in studying the effects of oxygen, the sensitivity of the receptor or process matters; for example even within the experimental results of C&D the effects of oxygen on length, mass and condition of cod gave differing boundaries between normoxic and sub-lethal layers. Secondly, if the "normoxic" or well oxygenated layer is defined as the layer within which differences in oxygen have no detectable adverse effect then the boundary depends not only on what effects are being studied but also on how sensitive the tests for adverse effects are. This is analogous to the controversy over "thresholds" for effects of radioactivity or pollutants on human health. I do not question the utility of using such defined layers for general descriptive purposes, but I do question whether they are the best way of carrying out the analysis in this study. CHOL use the study of oxygen effects on growth and condition of cod by C&D in defining the boundary between the normoxic and sub-lethal layers. The assumption that Baltic Sea cod, which live in a very different temperature and salinity environment from the Gulf of St Lawrence, respond in the same way to low oxygen is acceptable, however it is not obvious how they calculated the boundary value (4ml/l) that they take from C&D. The relationship between oxygen and condition factor in Figure 1c of C&D shows the "critical level of dissolved oxygen" i.e. the lower boundary of normoxia at 73% saturation. At the temperature and salinity of the experiments (10°C

and 28) this corresponds to 4.822ml/l of oxygen, not 4ml/l. An earlier paper (Casini et al., 2016) cited the same experimental source but said it "showed a significant decrease in condition already at 3 ml/l". To put these differences in context, the level of oxygen in the Bornholm Basin drops by 1ml/l every 10m depth increase; a 1ml/l drop in oxygen caused a roughly 10% drop in food intake in the experiments by C&D. Moving the boundary between the normoxic and sub-lethal layer up to 4.8ml/l will affect the correlations between oxygen and condition shown in CHOL, quite likely strengthening them. The task of defining boundaries between layers disappears if oxygen is treated as a continuous variable with progressive effects. The Abstract and the final paragraph of the Introduction of CHOL give the impression that the otolith microchemistry work was carried out as part of this paper; they should make it clear that all the results were published previously. Figure 6 in CHOL presents the same information as Figure 2 of Limburg and Casini 2019. Unresolved questions and other possible causes of declining condition of cod Why are cod moving into deeper water and thereby subjecting themselves to lower oxygen conditions? The likeliest explanation is that there is a life-history gain sufficient to offset the harmful effect of low oxygen on growth and condition. If prey availability is greater in deeper water, then even if it is not sufficient to prevent the observed decline in condition, it may still be better than their condition would have been if they had maintained their previous shallower depth distribution. An explanation put forward by CHOL (and others) is that cod avoid predation by seals and cormorants by moving into deeper water. Predators would thereby be having an indirect effect – not eating the cod, but causing their growth and condition to decline. This explanation may be valid, but is hard to test. CHOL and others (e.g. Neunfeldt 2019, 2020) have proposed that observed decline in stomach fullness, energy intake and the resultant effects on growth and condition of small cod may be due to decline in benthic prey, but they cite no direct evidence of decline in benthos, therefore this remains an untested hypothesis. Neunfeldt 2020 argue that the decline of benthic prey in cod stomachs is evidence that the availability of benthos has declined, but this seems like circular reasoning. Direct evidence of declining populations of benthos would be more convincing,

especially since our basic physiological knowledge and experimental results tell us that during current low oxygen conditions cod will eat less. Neunfeldt 2020 also argue that cod are able to avoid the negative effects of declining oxygen by making frequent (diel or shorter) vertical migrations into well-oxygenated water. If this were the case then it should undermine the distributional and otolith microchemical relationships between decline in oxygen and cod condition shown by CHOL. A full exploration of the evidence for vertical migration needs to take into account the changing diet and depth distribution of different life stages of cod (pelagic and settling juveniles, small benthic feeding cod, larger pelagic feeding cod) and also the estuarine hydrographic strucuture of the Baltic Sea, with a warmer, saline bottom layer separated from a cooler, fresher upper layer by a permanent halocline and seasonal stratification of the upper layer. Neunfeldt 2020 cite a study of otolith opacity (Hüssy, 2010)† as showing that small cod migrate across thermal gradients, but the resolution of the otolith rings is insufficient to show that this is a regular, short term pattern of behaviour and vertical migration by benthic-feeding small cod that live on the seabed below the halocline would take them into colder water. It is only when they grow to over 25cm and begin to switch their diet to pelagic prey that small cod are caught well above the seabed, as shown by pelagic trawling (Figure 3c (Andersen, Lundgren, Neuenfeldt, & Beyer, 2017)†) This by no means exhausts the list of unresolved questions and possible explanations of declining condition in cod, but probably takes us as far as is justified in the context of a review of CHOL. Minor editorial comments line 23 and 62 -The expansion of hypoxic areas has been quite rapid, but not exponential line 76 – make it clear that this explanation is inference and not based on evidence line 178 "these" presumably refers to "large fish" - better to say so.

Brander, K. (2020). Reduced growth in Baltic Sea cod may be due to mild hypoxia. ICES Journal of Marine Science, (December 2019), 2019–2021. https://doi.org/10.1093/icesjms/fsaa041 Casini, M., Käll, F., Hansson, M., Plikshs, M., Baranova, T., Karlsson, O., . . . Hjelm, J. (2016). Hypoxic areas, density-dependence and food limitation drive the body condition of a heavily exploited marine

fish predator. Royal Society Open Science, 3(10). https://doi.org/10.1098/rsos.160416
Chabot, D., & Dutil, J. D. (1999). Reduced growth of Atlantic cod in non-lethal hypoxic conditions. Journal of Fish Biology, 55, 472–491. Hüssy, K. (2010). Why is age determination of Baltic cod (Gadus morhua) so difficult? ICES Journal of Marine Science, 67(6), 1198–1205. https://doi.org/10.1093/icesjms/fsq023 Limburg, K. E., & Casini, M. (2019). Otolith chemistry indicates recent worsened Baltic cod condition is linked to hypoxia exposure. Biology Letters, 15(12), 20190352. https://doi.org/10.1098/rsbl.2019.0352 Neuenfeldt, S., Bartolino, V., Orio, A., Andersen, K. H., Andersen, N. G., Niiranen, S., . . . Casini, M. (2019). Feeding and growth of Atlantic cod (Gadus morhua L.) in the Eastern Baltic Sea under environmental change. https://doi.org/10.1093/icesjms/fsz224 Neuenfeldt, S. et al (2020) Response to Brander (2020) in press ICES Journal of Marine Science

---

## Short Comment (SC1) · 9 May 2020

We believe this comment has the same content as the first comment (RC1) from the same Reviewer. Our reply was provided to RC1.

---

## Author Comment (AC1) · 9 May 2020

We thank the reviewer for his thorough comments.

In literature, there have been only two studies investigating the relation between Baltic deoxygenation and cod condition, i.e. Casini et al. (2016) and Limburg & Casini (2019). In the former paper, a strong correlation was found between the extent of hypoxic areas (defined in that paper as km2 with oxygen < 2 ml/l) and condition, but the mechanisms potentially explaining the statistical relationships were not investigated but just

proposed, i.e. decline in benthic food, changes in cod behavior/distribution, direct physiological stress, or of course a combination of these. In the second paper (Limburg & Casini 2019) it was shown that fish in low condition at capture were exposed during their lives to lower oxygen levels than those in good condition (at least from the mid-1990s), without saying anything about the distribution of the population, and therefore whether or not a large part of the population indeed experienced stressful circumstances. Therefore, the original triggers and the mechanisms relating hypoxia to the average Baltic cod condition in the population were indeed elusive (and we think they still need attention), as we state in the abstract of the new paper (referred to as CHOL, from the initial of the authors names, following the terminology of the Reviewer; we refer here to the Reviewer as KB).

The CHOL paper takes a further step, showing that the cod population went progressively deeper in autumn and this, concomitant with a shallowing of the low-oxygen layers, increasing the spatial overlap between cod distribution and low-oxygen waters, and thus generating stressful circumstances for the cod population (exposure to waters with oxygen < 4 ml/l, detrimental for cod condition as found in experiments by Chabot & Dutil, 1999) (see below about the choice of the oxygen sub-lethal threshold in the CHOL paper). We finally showed that this increased overlap relates statistically to the decline in the mean population condition and to the proportion of fish with very low condition, both for juveniles and large fish. Therefore, the CHOL paper shows the original processes (deepening of the cod population concurrent with the shallowing of low-oxygen layers) creating the stressful circumstances relating to a decline in condition, for both small and large cod. In our opinion, this is a very important step forward in the understanding of the link between low-oxygen and cod condition, and in general for understanding the causes of the declined cod condition. Additionally, it is not so obvious that condition has to be directly linked to a general deoxygenation phenomenon, since mobile fish can change their distribution in response to that, as done by other fish species in other areas. This did not happen for the Baltic cod (conversely it went deeper, in autumn), and we think that finding the answer to why this has happened is

one of the next challenges for the scientists. Cod prey should also suffer from deoxygenation, although some are more tolerant to low oxygen; therefore, the question of why cod went deeper is not so trivial in our opinion and should be investigated as we suggested for future studies.

We are therefore totally in line with KB about the fact that "the issues to resolve are firstly whether cod redistribute themselves to remain in areas and depths with sufficient oxygen and if not then secondly whether the magnitude of ambient oxygen decline that cod experience is sufficient to explain all or only part of the observed change in their condition." This is exactly what we have done in the paper for both small and large cod in autumn. In addition, we have also investigated the original reasons creating these circumstances (i.e. both deepening of the population and shallowing of the low-oxygen layers), as well as estimated the overlap with the low-oxygen layers, known to affect cod condition, and estimated the relation between this overlap, the mean population condition and the percentage of fish with very low condition, for both small and large cod. This does not mean that direct exposure to low-oxygen is the sole driver of condition (even if oxygen decline is sufficient to explain a large part of the decline in condition), because there can be other contributing drivers and/or drivers that have co-varied with deoxygenation (food availability, parasites, inter- and intra-specific competition, etc...) that could also explain the reduced feeding level (see below). That is why further work is needed here too.

In the CHOL paper, we used 4 ml/l as sub-lethal oxygen threshold impairing cod condition. As KB correctly stated, 73% oxygen saturation (sub-lethal threshold in Chabot & Dutil (1999)) corresponds to 4.8 ml/l at the experimental conditions, but 65% oxygen saturation is the level from which the decline in condition was significant in Chabot & Dutil (1999) experiment, corresponding to 4.3 ml/l. We can therefore use either 4.3 or 4.8 ml/l in the revised version of the paper, to improve our analyses. We agree with KB that the real oxygen levels experienced by cod would be informative, likely even better, so in the revised paper we will be showing also the oxygen levels corresponding to the

annual depth distribution of cod in autumn, both for small and large cod. However, we think that the information about the shallowing of the 1 ml/l and 4 ml/l (the latter will be 4.3 or 4.8 ml/l in the revised paper) depths enriches the story, and together with the deepening of the cod distribution depth, it visually delivers a very clear message. Therefore, we would prefer to retain this.

The otolith analysis was already published in Limburg and Casini (2019), but the analysis was re-arranged as a new figure in CHOL. We thought that this was a nice conclusion of the story, but if the Reviewer and/or Editor prefer, we could either specify better that the idea was already presented previously, or delete the last figure and explain in the text the link between our results and those of Limburg and Casini (2019).

Exploring mechanistic relationships would need experimental setups. Using time-series, the statistical relationships have to be interpreted in light of what is known about the biology and ecology of the fish. In our case, we used the experimental results from Chabot & Dutil (1999) to relate the distribution of the cod population with the oxygen levels resulted to affect cod in experimental setups, and in the revised paper we will use more information coming from stomach content analysis (see below).

We agree with KB that Neuenfeldt et al. (2020) is an extremely important paper, showing that the lower energy intake observed in cod (using stomach content time-series) would predict a decrease growth in length that could explain the shift in size distribution of cod population towards lower sizes. The lower amount of benthos and pelagic fish in the diet of cod could be due to a decline in their availability (as suggested in Neuenfeldt et al. 2020) but also to a decline in appetite due to low-oxygen exposure (Chabot & Dutil 1999, Brander 2020) or other low oxygen-related physiological stress. Food intake can surely be the main driver of growth, but other factors can cause fish to allocate more energy to basic metabolism, reproduction etc. . . in some circumstances. For example, currently Baltic cod reproduce at a smaller size (around 20 cm) than before (30-35 cm) and this could mean a lower allocation of energy to growth and therefore also explain the growth decline. We agree with KB that such reasoning produces the

egg-chicken problem, but it brings us outside the scope of the CHOL paper.

In our analyses we investigate fish condition, not growth in length, and since the two traits are different (fish can grow fast in length, utilizing the stored energy reserves, but this at the detriment of condition, that is a ratio between weight and length) we do not want to mix them, and moreover the link between condition and growth has not been well established to our knowledge. However, in the revised paper, we will add more discussion about the decline in feeding level found in Neuenfeldt et al. (2020) that could link the increased exposure to low-oxygen levels to declined condition. However, there are some aspects that make this link not as straight forward as it seems. Neuenfeldt et al. (2020) show that feeding level has not declined for large cod, but the observed decline in condition has been more severe for large cod (Casini et al. 2016 and the new CHOL paper), suggesting perhaps that feeding level is not the sole driver of large cod condition and that therefore low oxygen has impacted cod condition also through different mechanisms, other than food intake. For example, large cod could experience shortage of benthic prey and therefore, proportionally, could be forced to eat more pelagic fish that require higher energy to catch. Moreover, cod was not in low-oxygen conditions before the early 1990s (see our CHOL paper), but the feeding level was already low (Neuenfeldt et al. 2020; see also ICES 2016), and so was condition (Casini et al. 2016, new CHOL paper), indicating that direct exposure to hypoxia is not always the driver of feeding level and condition (matching therefore with the results from otolith analyses in Limburg & Casini (2019)). In the revised paper, we will however discuss more our results to Neuenfeldt et al. (2020)'s findings about feeding levels, to link the increased overlap with low-oxygen waters to feeding level and condition after the early 1990s. We will moreover discuss more the CHOL paper results in view of Brander (2020) paper recently published.

References

Brander, K (2020). Reduced growth in Baltic Sea cod may be due to mild hypoxia. ICES J. Mar. Sci., doi:10.1093/icesjms/fsaa041.

Casini, M., Käll, F., Hansson, M., Plikshs, M., Baranova, T., Karlsson, O., Lundström, K., Neuenfeldt, S., Gårdmark, G., and Hjelm J. 2016. Hypoxic areas, density dependence and food limitation drive the body condition of a heavily exploited marine fish predator. R. Soc. Open Sci., 3, 160416. Doi: 10.1098/rsos.160416.

Chabot, D., and Dutil, J.-D. 1999. Reduced growth of Atlantic cod in non-lethal hypoxic conditions. J. Fish Biol., 55, 472–491.

ICES (2016). Report of the Workshop on Spatial Analyses for the Baltic Sea (WKSPATIAL), 3-6 November 2015, Rome, Italy. ICES CM 2015/SSGIEA:13. 37 pp.

Limburg, K., and Casini, M. 2019. Otolith chemistry indicates recent worsened Baltic cod condition is linked to hypoxia exposure. Biol. Lett., 15, 20190352.

Neuenfeldt, S., Bartolino, V., Orio, A., Andersen, K. H., Andersen, N. G., Niiranen, S., Bergström, U., Ustups, D. Kallasvuo, M., Kulatska, N., and Casini, M. 2020. Feeding and growth of Atlantic cod (Gadus morhua L.) in the Eastern Baltic Sea under environmental change. ICES J. Mar. Sci., 77: 624–632.

---

## Referee Comment (RC3) · Anonymous Referee #2 · 10 May 2020

The study examines data sets of oxygen concentration, cod catch across sites and depth and cod condition factor in the Baltic sea across forty years. During this period hypoxic areas have increased, and cod body condition has decreased. Importantly cod are increasingly found in low oxygen waters. An important finding is that lifetime exposure to low oxygen correlates to body condition on an individual level.

The paper is interesting, and the patterns are convincing. Inevitable any conclusions drawn from parallel changes in two or more metrics without a test will be speculative. Nevertheless, I think the authors do a good enough job of highlighting hypoxia as a contributor to decreasing cod condition. However, I think the description on confounding effects and other contribution factors could be improved. For example, although hypoxia may well contribute greatly to low growth of cod in the current system the drivers of a decrease in condition are the triggers of a change in depth distribution and the cause of low oxygen.

Furthermore, there is no description of any statistical analysis. Mostly the patterns are "analyzed" by eye and described in the results chapter (related note referring to figures as you describe results). This approach may occasionally be valid – and the patterns described are convincing enough - but at least some sort of quantification of the size of effects across time should use when describing them (reduced from x to x). A statistic test is used for the otolith data, but this is not included in the methods.

The results from the otolith analysis is interesting yet this part of the paper is referred to as an afterthought throughout the paper. I think this analysis warrants increased value, both by adding to the introduction enough background material to allow readers to evaluate the validity of the methods on know of any prior findings and in the a fuller description of methods including how the otoliths were selected.

26+28: Is "processes" the right word? 100: What is the sample size? 101: is this data stable once entered, or is it subject to change? In the last case, a date of retrieval would be handy to include. 105: there are different ways to measure 'total length', maybe explain in more detail how it was done in this study. 107: why is SD26-28 chosen and not for example not 29? 108/109: why is the subdivision of big and small cod made and why those specific lengths? What happens with fish between 29 and 40 cm? 109: Quarter 4 also includes part of the winter. Why not mentioning the exact months instead of season or quarter 4? 117: why are those class divisions different from row 108? 135: it is later explained, but I would rather put here the <0.8 (Eero et al 2012), explaining the 'very low' con-dition 160-191: I see many statements as 'more' and 'lower' and 'deeper', but it is very descriptive, and I miss actual numbers in some places and statistical tests to prove these statements. Also, how many data points were

retrieved, how big was the sample size? 171: which depth? 186: The oxygen layers are almost the same, but not totally. I understand this is because they are weighed with the SD-specific distribution of the cod, but I think it makes things clearer if you write somewhere that this means that it differs between the big and the small cod (it took me a while to understand). 267: I miss a note about that it is not 100% sure that the cod are actually in those low oxygen waters, because that was not directly measured. However, the additional otolith results make it very plausible that this is the case. 273: Was there a way to directly link otolith chemistry with body condition? (e.g. from the same individual?) Why do you think the overlap between cod and oxygen layers is oscillating? (why is the oxygen stratification oscillating?) 475/476/481/486: you use here the whole word 'subdivision', while in the previous description (472) you al-ready used SD 490 post-2000? This is differently described throughout the text.

Figure 3: Is there a possible explanation for the high condition in 1996 in SD25

Figure 6: 2000 onward is called 'post 2000' in the text. Why are there squares in the boxes

---

## Author Comment (AC2) · 12 May 2020

We thank the reviewer for the helpful comments.

The reviewer asked about other factors that drive the poor condition of cod. In fact, we presented briefly the potential other factors contributing to the cod condition patterns in the Introduction to provide some background, specifying that in literature deoxygenation has been advocated as one of the major drivers of the condition decline (e.g. Casini et al. 2016). However, in the present manuscript, we specifically wanted to ex-

plore the link between deoxygenation and condition and investigate the circumstances that brought cod into greater contact with (and higher exposure to) low-oxygen waters. We appreciate the reviewer's point, and thus we can add some text in the Discussion section about the alternative factors that could contribute to the patterns in cod condition, especially before the mid 1990s when the cod population seemed outside the sub-lethal oxygen layers but condition was quite low. We can also add some sentences about the reasons of the shallowing of the low-oxygen layers, from literature. On the other hand, the reasons for the deepening of the cod population have not been investigated, only speculated about in other papers (Orio et al. 2019). These are beyond the scope of our paper, but we have suggested that this is an important question to answer in futures studies.

In the revised manuscript we will be more quantitative, spelling out the most important changes across time. We will provide information on the statistics that have been used, i.e. linear regressions for the time-series of condition and overlap to low-oxygen layers, and Anova for the otolith analysis. The otolith analysis is a modification of the analysis done in Limburg & Casini (2019), so we thought not to explain the method too much in detail but just refer to that paper for further information for example about the sampling etc., but we could do that if the Reviewer #2 and/or the Editor prefer so.

In the revision we will also address the specific questions and comments raised by the Reviewer #2. The main responses are listed below:

- (Q on line 107) We did not use Subdivisions 29 or northward because of the spatial distribution of the cod population. Since the early 1990s the population has concentrated in the southern Baltic Sea.

- (Q on lines 108-109) The two length groups for condition were selected to represent small and large fish. The small fish can also be seen as juveniles even though the size at maturity has declined with time for this population. The large fish on the other hand can all be considered adults. Currently, there are very few cod above 50 cm and

therefore we could not use larger size-classes.

- (Q on line 117) The population distributions, divided in < 30cm and ≥ 30 cm, come from Orio et al. (2019). In the condition estimations, we did not want to use too large ranges of fish sizes in one group because Fulton condition factor (used in the paper we refer to and compare to ours) can be affected by fish size. Moreover, cod start to become piscivorous around 30 cm and therefore fish below 30 cm (but larger than in the plankton- and nektobenthos-feeder phase, around 15 cm) can be considered occupying similar ecological niche. Therefore, the 20-29 and 40-49 size groups were chosen for condition estimation just to represent the small and large sizes with different ecological niches and therefore likely different behavior and food requirements.

- (Q on line 273) Regarding the question "Was there a way to directly link otolith chemistry with body condition? (e.g. from the same individual?)". Yes, it is possible analyzing the Mn/Mg elements ratio in the otoliths of individual fish, see Limburg & Casini (2018, 2019).

References

Casini, M., Käll, F., Hansson, M., Plikshs, M., Baranova, T., Karlsson, O., Lundström, K., Neuenfeldt, S., Gårdmark, G. and Hjelm J. 2016. Hypoxic areas, density dependence and food limitation drive the body condition of a heavily exploited marine fish predator. R. Soc. Open Sci., 3, 160416. Doi: 10.1098/rsos.160416.

Limburg, K.E and Casini, M. 2019. Otolith chemistry indicates recent worsened Baltic cod condition is linked to hypoxia exposure. Biol. Lett., 15, 20190352.

Limburg, K.E. and Casini, M. 2018. Effect of marine hypoxia on Baltic Sea cod Gadus morhua: evidence from otolith chemical proxies. Frontiers in Marine Science, 5: 482.

Orio, A., Bergström, U., Florin, A.-B., Lehmann, A., Šics, I. and Casini, M. 2019. Spatial contraction of demersal fish populations in a large marine ecosystem. Journal of Biogeography, 46: 633-645.

---

## Referee Comment (RC4) · Jan Dierking (Referee) · 21 May 2020

General comments

The manuscript by Casini et al. aims to shed new light on the mechanisms linking the expansion of oxygen minimum zones in the Baltic Sea and the decline in condition of Baltic cod since the 1990s. The manuscript therefore addresses a topic of global interest, the impact of environmental changes and in particular deteriorating oxygen conditions on the status of fish stocks. Moreover, the condition decline of cod has received

a lot of scientific attention in the Baltic region, and also has applied consequences in affecting mortality and economic value of cod. There has been a large number of publications directly or indirectly addressing the potential mechanisms driving the condition decline, including previous work linking low condition of cod to exposure to low oxygen conditions (Limburg and Casini 2019). At the same time, the implication and relative importance of different mechanisms is still incompletely understood and requires further attention. Since the Baltic has seen very pronounced environmental impacts, including the strong expansion of oxygen minimum zones, it can be considered as model for other coastal areas, and insights gained from the cod story can also be of value for fish stocks under environmental stressors elsewhere. The manuscript therefore addresses questions that are relevant on both the regional Baltic scale and of interest to a general (global) readership.

In terms of the specific analyses, I find the combination of time series on cod depth distribution and on the depth layers at which waters with low oxygen concentrations are present, the resulting calculation of the overlap between these areas, and finally the correlation of overlap with cod condition interesting and relevant. The study provides an independent confirmation of previous results by Limburg and Casini (2019) that implicate exposure to waters with low oxygen concentrations in low cod condition, and provides an important new perspective on the nature of that exposure. This leads to interesting new questions, in particular why the cod depth distribution moves deeper into low oxygen waters in the fall, instead of shallower to avoid such layers.

At the same time, I see a number of major limitations with the present version of the manuscript that should be addressed. I will list these in the following:

—Tendency to oversell the results: this is recurring through the title, abstract and discussion. For example, the study title is not in line with the results. The title implies that the study results alone explain the low condition of Baltic cod, when in reality, the study sheds additional light on one potential mechanism, direct exposure to low oxygen waters, which does not rule out alternative mechanisms (both linked to expanding

oxygen minimum zones and to other factors) that have been proposed before. Abstract L27-29: point out more clearly that the study is assessing the role of direct exposure to low oxygen waters, not "the processes". Discussion L223: should be "one mechanism" not "the mechanisms". Conclusion L293: should be "shown here one mechanism", not "the mechanisms".

—Delineation of results from previous work: Exposure to low oxygen water was already previously linked to the Baltic cod condition decline by Limburg and Casini (2019) using otolith microchemistry. This study is cited and referred to by Casini et al., but still, the apparent narrative here is that the exposure to low oxygen waters is shown via the identification of increasing overlap of the depths of low oxygen waters and the cod depth distribution, and that this is then confirmed with otolith microchemistry in this manuscript (e.g., abstract LL29-34, Introduction LL89-93, Discussion LL224-228). This really has it backwards. I suggest to instead clearly lay out key results and conclusions from Limburg and Casini 2019 in the Introduction, and then use this as rationale for the (relevant and interesting) independent confirmation and new insights into the specific patterns of exposure to low oxygen waters in this manuscript.

—Use and presentation of otolith microchemistry dataset from Limburg and Casini 2019 in this study (connected to previous comment): I would strongly recommend the exclusion of these data from the present manuscript. To me, the analysis and results mirror the previous publication by Limburg and Casini too closely to warrant inclusion here. The authors acknowledge the previous study, but without going into details. However, the data set, analyses, discussion points (Section 4.2) and conclusions are largely the same. Also, the results from otolith microchemistry analyses are not formally correlated to the depth distribution analyses, and appear rather like an "afterthought" in this manuscript. The inclusion in the manuscript thus unnecessarily duplicates previous work. If conclusions from the previous work are instead clearly presented in the Introduction, this will provide the rationale for the real strength and novelty of the present study, the depth distribution analyses. New insights from this independent approach

compared to the insights from the original otolith microchemistry approach could then also be discussed more explicitly in the Discussion. Interestingly, all conclusions in the conclusion section of the manuscript (LL293-306) relate to this aspect of the study anyway.

—Statistical analyses: Right now, the manuscript is lacking in formal statistical assessments. This includes statistical approaches to assess the significance and nature of temporal trends in the depths of low oxygen waters, cod depth distributions and overlap, as well as the formal assessment of the link of overlap and cod condition over time. The Material and Methods should then also include a dedicated section outlining statistical approaches. In this context, looking at Figure 4 of the manuscript, many of the observed temporal changes do not look linear. E.g., for SD26-28, cod mean depth was essentially stable after 1990, and for SD25, neither cod depth distribution nor depth of low oxygen water appears to change significantly between 2008 and 2018. Formal statistical analysis would therefore have the potential to lead to additional insights beyond the points included in the manuscript.

Specific comments

Throughout the entire manuscript, I was waiting for an explanation for the discrepancy of the cod depth distribution trends over time between the very similar data sets and analyses in Orio et al. 2019 (showing cod distributions at least for SD26-28 becoming shallower since the 1990s) and this manuscript. This was then given in the second to last sentence of the conclusions :) I suggest to explicitly explain the difference between the datasets (fall versus other seasons) already in the Material and Methods, and then discuss this interesting difference between seasons in the main part of the Discussion, not just in the Conclusion.

L53: Would cite Chabot and Dutil 1999 here already.

L60: Suggest addition of Reusch et al 2018 as probably best reference for combined strong temporal changes in temperature, eutrophication, oxygen in the Baltic Sea.

L60-61: to my knowledge, the degradation of benthic communities is NOT well documented in the Baltic Sea, and lack of time series on benthic communities has been one of the issues hampering understanding of consequences of expanding oxygen minimum ones. Rephrase.

L71: see major comments regarding previous results from Limburg and Casini 2019. Suggest to present in much more depth here and explain that link between low condition and exposure to low oxygen water was established in that study.

L73: suggest to mention the actual mechanism connected to this, density dependence.

L73-75: add mechanism proposed by Brandner 2020, mild hypoxia reducing rate of digestion.

L92-96: The otolith works comes in like an afterthought here, since it is not set up in any way in the Introduction section (linked to major comment regarding otolith work)

Section 2.1: more clearly point out that this (or very similar) cod condition time series were previously published and are here updated to 2018?

LL107-109: please explain rationale of using size class 20-29 and 40-49 cm for condition calculations.

Section 2.2: suggest to point out more clearly the key difference between studies, focus on fall here versus all seasons in Orio et al 2019 (see my previous comment above).

LL125-135: I am not a physiologist, but I guess in principle use of oxygen as continuous variable (instead of somewhat arbitrary boundaries) would make sense. I can see that use of specific limits facilitates analysis, but would mention this possible limitation.

L155: Explain the rationale of using a Fulton's k of 0.9. Also give other thresholds (e.g., "very low" used later in L163) here already.

Section 3.2: in the Discussion section (not here), suggest to discuss the patterns observed for fall here compared to the patterns in Orio et al 2019 reporting cod depth

distribution contraction to shallower water for SD26-28 when looking at the entire year.

Section 3.2, 3.3, 3.4: would all benefit a lot from formal statistics.

L243: I think the discussion of mechanisms that can explain what drives cod into layers with low oxygen levels is quite central, since it relates to the key novel finding of this manuscript. Suggest to therefore not state that "beyond scope" of manuscript, but rather state that you can only speculate and will discuss possible causes as systematically as possible.

LL244-245: The role of temperature was also the first thing that came to my mind, but I then wondered about actual temperature profiles in fall, and whether they would support these considerations. It would be useful to include information on prevailing temperature depth profiles in fall as background for the discussion.

L283: Should read "although we have confirmed here that …" and refer to Limburg and Casini 2019.

LL283-291: Discussion of other factors could be more extensive. Cite Brander et al 2020 here as well.

L297: Agree, very interesting future direction, and a question that really results for the first time from the analyses in this manuscript (not possible from Limburg and Casini 2019) – this would be worth pointing out.

Figures: I suggest to add a figure to illustrate key findings regarding the correlation of cod condition and the overlap of cod depth distribution and low oxygen waters.

Related to the general comment regarding the presentation of otolith microchemistry data in this manuscript, Figure 6 of this manuscript appear to be an alternative view of Figure 2 c in Limburg and Casini 2019, i.e., not adding new information here that could not be provided from that manuscript.

Technical corrections

LL23-24, L62: wording should be more precise – "exponential increase" not really correct, suggest "strong increase"; "largest marine dead zone", unnecessarily dramatic.

LL26: "elusive" does not really reflect that specific alternative mechanisms have been proposed.

LL29-32: rephrase, confusing wording.

L59-60: Wording in Breitburg et al 2018 is more scientific ("low O2 areas have become more extensive and severe") – suggest to follow this approach.

L81: start new paragraph, focusing on effects and not mechanisms from here on.

L82-82: rephrase "lamented"

L194: "in a couple…" – word missing?

L254: "hostile waters" – suggest to rephrase

References

Brander K (2020) Reduced growth in Baltic Sea cod may be due to mild hypoxia. ICES Journal of Marine Science. DOI: 10.1093/icesjms/fsaa041

Breitburg D, Levin LA, Oschlies A, Gregoire M, Chavez FP, Conley DJ, Garcon V, Gilbert D, Gutierrez D, Isensee K, Jacinto GS, Limburg KE, Montes I, Naqvi SWA, Pitcher GC, Rabalais NN, Roman MR, Rose KA, Seibel BA, Telszewski M, Yasuhara M, Zhang J (2018) Declining oxygen in the global ocean and coastal waters. Science 359:eaam7240. 10.1126/science.aam7240

Chabot D, Dutil J-D (1999) Reduced growth of Atlantic cod in non-lethal hypoxic conditions. Journal of Fish Biology 55:472-491. DOI: 10.1111/j.1095-8649.1999.tb00693.x

Limburg KE, Casini M (2019) Otolith chemistry indicates recent worsened Baltic cod condition is linked to hypoxia exposure. Biology Letters 15:20190352. DOI:10.1098/rsbl.2019.0352

[Figure]

Orio A, Bergström U, Florin A-B, Lehmann A, Šics I, Casini M (2019) Spatial contraction of demersal fish populations in a large marine ecosystem. Journal of Biogeography 46:633-645. DOI: 10.1111/jbi.13510

Reusch TBH, Dierking J, Andersson HC, Bonsdorff E, Carstensen J, Casini M, Czajkowski M, Hasler B, Hinsby K, Hyytiäinen K, Johannesson K, Jomaa S, Jormalainen V, Kuosa H, Kurland S, Laikre L, MacKenzie BR, Margonski P, Melzner F, Oesterwind D, Ojaveer H, Refsgaard JC, Sandström A, Schwarz G, Tonderski K, Winder M, Zandersen M (2018) The Baltic Sea as a time machine for the future coastal ocean. Science Advances 4. DOI: 10.1126/sciadv.aar8195

---

## Author Comment (AC4) · 10 Jun 2020

We thank the reviewer for the helpful comments.

Reply to the general comments:

- The paper is interesting, and the patterns are convincing. Inevitable any conclusions drawn from parallel changes in two or more metrics without a test will be speculative. Nevertheless, I think the authors do a good enough job of highlighting hypoxia as a contributor to decreasing cod condition. However, I think the description on confounding

effects and other contribution factors could be improved. For example, although hypoxia may well contribute greatly to low growth of cod in the current system the drivers of a decrease in condition are the triggers of a change in depth distribution and the cause of low oxygen.

The reviewer asked about other factors that drive the poor condition of cod. In fact, we presented briefly the potential other factors contributing to the cod condition patterns in the Introduction to provide some background, specifying that in literature deoxygenation has been advocated as one of the major drivers of the condition decline (e.g. Casini et al. 2016). Our present manuscript is focusing on showing the processes (deepening of cod population and shallowing of low-oxygen layers) explaining the link between the general Baltic deoxygenation and condition (as shown by Casini et al. (2016)) and putting in a population context what found previously in the cod otoliths by Limburg & Casini (2019). We appreciate the reviewer's point, and thus we can add some text in the Discussion section about the alternative factors that could contribute to the patterns in cod condition, especially before the mid 1990s when the cod population seemed outside the sub-lethal oxygen layers but condition was quite low. We can also add some sentences about the reasons of the shallowing of the low-oxygen layers, from literature. On the other hand, the reasons for the deepening of the cod population have not been investigated, only speculated about in other papers (Orio et al. 2019). These are beyond the scope of our paper, but we have suggested that this is an important question to answer in futures studies.

- Furthermore, there is no description of any statistical analysis. Mostly the patterns are "analyzed" by eye and described in the results chapter (related note referring to figures as you describe results). This approach may occasionally be valid – and the patterns described are convincing enough - but at least some sort of quantification of the size of effects across time should use when describing them (reduced from x to x). A statistic test is used for the otolith data, but this is not included in the methods. The results from the otolith analysis is interesting yet this part of the paper is referred

to as an afterthought throughout the paper. I think this analysis warrants increased value, both by adding to the introduction enough background material to allow readers to evaluate the validity of the methods on know of any prior fidings and in the a fuller description of methods including how the otoliths were selected.

In the revised manuscript we will be more quantitative, spelling out the most important changes across time. We now also estimated the actual oxygen that the population has been experiencing over time (not only the overlap with low-oxygen levels below a certain threshold) and we perform statistical analysis relating this with fish condition. The otolith analysis is a modification of the analysis done in Limburg & Casini (2019), so we thought not to explain the method too much in detail but just refer to that paper for further information for example about the sampling. However, we have now opted to delete the part on otolith chemistry form the main body of the text but add it in Supplementary material.

Reply to the specific comments:

- 26+28: Is "processes" the right word?

We believe yes, we could say also "mechanisms".

- 100: What is the sample size?

We have added it.

- 101: is this data stable once entered, or is it subject to change? In the last case, a date of retrieval would be handy to include.

We have now added the date of data extraction for the years after 1990, which can undergo slight updates in the ICES DATRAS database. The years before 1990s are from historical databases and therefore not subject to change.

- 105: there are different ways to measure 'total length', maybe explain in more detail how it was done in this study.

[Figure]

Done.

- 107: why is SD26-28 chosen and not for example not 29?

We have now explained the reason.

- 108/109: why is the subdivision of big and small cod made and why those specific lengths? What happens with fish between 29 and 40 cm?

The two length groups for condition were selected to represent small and large fish, as stated in the paper. The small fish can also be seen as juveniles even though the size at maturity has declined with time for this population. The large fish on the other hand can all be considered adults. Currently, there are very few cod above 50 cm and therefore we could not use larger size-classes. We have now edited a little this part.

- 109: Quarter 4 also includes part of the winter. Why not mentioning the exact months instead of season or quarter 4?

Done

- 117: why are those class divisions different from row 108?

The population distributions, divided in $< 30cm$ and $\geq 30$ cm, come from Orio et al. (2019). In the condition estimations, we did not want to use too large ranges of fish sizes in one group because Fulton condition factor (used in the paper we refer to and compare to ours) can be affected by fish size. Moreover, cod start to become piscivorous around 30 cm and therefore fish below 30 cm (but larger than in the plankton- and nektobenthos-feeder phase, around 15 cm) can be considered occupying similar ecological niche. Therefore, the 20-29 and 40-49 size groups were chosen for condition estimation just to represent the small and large sizes with different ecological niches and therefore likely different behavior and food requirements. We have now added information into this part.

- 135: it is later explained, but I would rather put here the <0.8 (Eero et al 2012),

explaining the 'very low' condition

Done

- 160-191: I see many statements as 'more' and 'lower' and 'deeper', but it is very descriptive, and I miss actual numbers in some places and statistical tests to prove these statements. Also, how many data points were retrieved, how big was the sample size?

We do not think we need statistical tests to explain the long-term patterns, what is important is the overlap between the cod population and low-oxygen layers. However, we tried to add some more quantitative information in the text and not only percentages. We now also estimated the actual oxygen that the population has been experiencing over time (not only the overlap with low-oxygen levels below a certain threshold) and we perform statistical analysis relating this with fish condition. We have also added the samples sizes in the Methods.

- 171: which depth?

Done, we have improved this description.

- 186: The oxygen layers are almost the same, but not totally. I understand this is because they are weighed with the SD-specific distribution of the cod, but I think it makes things clearer if you write somewhere that this means that it differs between the big and the small cod (it took me a while to understand).

Done

- 267: I miss a note about that it is not 100% sure that the cod are actually in those low oxygen waters, because that was not directly measured. However, the additional otolith results make it very plausible that this is the case.

We agree, fish can move and therefore we cannot be sure that those with very low condition spent most of their time in low-oxygen waters (even if they were caught there)

from the time-series, but as the Reviewer #2 also says, this is very plausible also considering the otoliths' analyses in Limburg and Casini (2019) (and the modified analyses now in Supplement).

- 273: Was there a way to directly link otolith chemistry with body condition? (e.g. from the same individual?) Why do you think the overlap between cod and oxygen layers is oscillating? (why is the oxygen stratification oscillating?)

Yes, it is possible analyzing the Mn/Mg elements ratio in the otoliths of individual fish, see Limburg & Casini (2018, 2019).

- 475/476/481/486: you use here the whole word 'subdivision', while in the previous description (472) you al-ready used SD

We have edited this to be more consistent.

- 490 post-2000? This is differently described throughout the text.

We have edited this to be more consistent.

- Figure 3: Is there a possible explanation for the high condition in 1996 in SD25

In general, the mid 1990s are characterized by good oxygen conditions (low extent of hypoxic areas) and a large increase in the sprat stock, probably boosting condition. We feel that going into these details bring us out of the paper's scope and we prefer not to focus on single annual values but on the general patterns.

- Figure 6: 2000 onward is called 'post 2000' in the text. Why are there squares in the boxes

We are now more consistent in the terminology. In the caption we have now also specified what the squares and the boxes are.

References

Casini, M., Käll, F., Hansson, M., Plikshs, M., Baranova, T., Karlsson, O., Lundström,

K., Neuenfeldt, S., Gårdmark, G. and Hjelm J. 2016. Hypoxic areas, density dependence and food limitation drive the body condition of a heavily exploited marine fish predator. R. Soc. Open Sci., 3, 160416. Doi: 10.1098/rsos.160416.

Limburg, K.E and Casini, M. 2019. Otolith chemistry indicates recent worsened Baltic cod condition is linked to hypoxia exposure. Biol. Lett., 15, 20190352.

Limburg, K.E. and Casini, M. 2018. Effect of marine hypoxia on Baltic Sea cod Gadus morhua: evidence from otolith chemical proxies. Frontiers in Marine Science, 5: 482.

Orio, A., Bergström, U., Florin, A.-B., Lehmann, A., Šics, I. and Casini, M. 2019. Spatial contraction of demersal fish populations in a large marine ecosystem. Journal of Biogeography, 46: 633-645.

---

## Author Comment (AC5) · 10 Jun 2020

We thank the reviewer for the helpful comments.

Reply to the general comments:

- Tendency to oversell the results: this is recurring through the title, abstract and discussion. For example, the study title is not in line with the results. The title implies that the study results alone explain the low condition of Baltic cod, when in reality, the study sheds additional light on one potential mechanism, direct exposure to low oxygen waters, which does not rule out alternative mechanisms (both linked to expanding oxygen minimum zones and to other factors) that have been proposed before. Abstract L27-29: point out more clearly that the study is assessing the role of direct exposure to low oxygen waters, not "the processes". Discussion L223: should be "one mechanism" not "the mechanisms". Conclusion L293: should be "shown here one mechanism", not "the mechanisms".

We have now gone through the manuscript and edited some sentences not to oversell our results.

- Delineation of results from previous work: Exposure to low oxygen water was already previously linked to the Baltic cod condition decline by Limburg and Casini (2019) using otolith microchemistry. This study is cited and referred to by Casini et al., but still, the apparent narrative here is that the exposure to low oxygen waters is shown via the identification of increasing overlap of the depths of low oxygen waters and the cod depth distribution, and that this is then confirmed with otolith microchemistry in this manuscript (e.g., abstract LL 29-34, Introduction LL 89-93, Discussion LL 224-228). This really has it backwards. I suggest to instead clearly lay out key results and conclusions from Limburg and Casini 2019 in the Introduction, and then use this as rationale for the (relevant and interesting) independent confirmation and new insights into the specific patterns of exposure to low oxygen waters in this manuscript.

We have now followed the suggestion from the Reviewer, specifying in the Introduction that in Limburg & Casini (2019) it was shown that fish in low condition at capture were exposed during their lives to lower oxygen levels than those in good condition (at least from the mid-1990s), without saying anything about the distribution of the population, and therefore whether or not a large part of the population indeed experienced stressful circumstances, that could explain the low population condition found in Casini et al. (2016).

- Use and presentation of otolith microchemistry dataset from Limburg and Casini 2019

in this study (connected to previous comment): I would strongly recommend the exclusion of these data from the present manuscript. To me, the analysis and results mirror the previous publication by Limburg and Casini too closely to warrant inclusion here. The authors acknowledge the previous study, but without going into details. However, the dataset, analyses, discussion points (Section4.2) and conclusions are largely the same. Also, the results from otolith microchemistry analyses are not formally correlated to the depth distribution analyses, and appear rather like an "afterthought" in this manuscript. The inclusion in the manuscript thus unnecessarily duplicates previous work. If conclusions from the previous work are instead clearly presented in the Introduction, this will provide the rationale for the real strength and novelty of the present study, the depth distribution analyses. New insights from this independent approach compared to the insights from the original otolith microchemistry approach could then also be discussed more explicitly in the Discussion. Interestingly, all conclusions in the conclusion section of the manuscript (LL293-306) relate to this aspect of the study anyway.

We have now followed the suggestion from the Reviewer, specifying in the Introduction that in Limburg & Casini (2019) it was shown that fish in low condition at capture were exposed during their lives to lower oxygen levels than those in good condition (at least from the mid-1990s), without saying anything about the distribution of the population, and therefore whether or not a large part of the population indeed experienced stressful circumstances, that could explain the low population condition found in Casini et al. (2016). In the current submitted paper we however would like to show the otolith figure (that is a rearrangement and slightly different analysis of Limburg & Casini (2019)) and insert it in the Supplementary material explaining that it is a modification from Limburg & Casini (2019).

- Statistical analyses: Right now, the manuscript is lacking in formal statistical assessments. This includes statistical approaches to assess the significance and nature of temporal trends in the depths of low oxygen waters, cod depth distributions and overlap, as well as the formal assessment of the link of overlap and cod condition over time. The Material and Methods should then also include a dedicated section outlining statistical approaches. In this context, looking at Figure 4 of the manuscript, many of the observed temporal changes do not look linear. E.g., for SD26-28, cod mean depth was essentially stable after 1990, and for SD25, neither cod depth distribution nor depth of low oxygen water appears to change significantly between 2008 and 2018. Formal statistical analysis would therefore have the potential to lead to additional insights beyond the points included in the manuscript.

We agree that the trends of the depth patterns are not linear, that is also why a statistical tests of the temporal patterns would not provide much additional information in our opinion. We now estimated the actual oxygen that the population has been experiencing over time (not only the overlap with low-oxygen levels below a certain threshold) and we perform statistical analysis relating this with fish condition.

Reply to the specific comments:

- Throughout the entire manuscript, I was waiting for an explanation for the discrepancy of the cod depth distribution trends over time between the very similar data sets and analyses in Orio et al. 2019 (showing cod distributions at least for SD26-28 becoming shallower since the 1990s) and this manuscript. This was then given in the second to last sentence of the conclusions:) I suggest to explicitly explain the difference between the datasets (fall versus other seasons) already in the Material and Methods, and then discuss this interesting difference between seasons in the main part of the Discussion, not just in the Conclusion.

We present shortly a discussion of the differences about this already in the Discussion now (before the Conclusions).

- L53: Would cite Chabot and Dutil 1999 here already.

Done

- L60: Suggest addition of Reusch et al 2018 as probably best reference for combined strong temporal changes in temperature, eutrophication, oxygen in the Baltic Sea.

Done

- L60-61: to my knowledge, the degradation of benthic communities is NOT well documented in the Baltic Sea, and lack of time series on benthic communities has been one of the issues hampering understanding of consequences of expanding oxygen minimum ones. Rephrase.

Done

- L71: see major comments regarding previous results from Limburg and Casini 2019. Suggest to present in much more depth here and explain that link between low condition and exposure to low oxygen water was established in that study.

Done. We have now specified in the Introduction that in Limburg & Casini (2019) it was shown that fish in low condition at capture were exposed during their lives to lower oxygen levels than those in good condition (at least from the mid-1990s), without saying anything about the distribution of the population, and therefore whether or not a large part of the population indeed experienced stressful circumstances, that could explain the low population condition found in Casini et al. (2016).

- L73: suggest to mention the actual mechanism connected to this, density dependence.

Done, but we also meant change in the habitat occupation, not only contraction, we have now rephrased the sentence.

- L73-75: add mechanism proposed by Brandner 2020, mild hypoxia reducing rate of digestion.

The mechanism is already included in the sentence (stress due to hypoxia exposure), but we have clarified it adding the reference.

- L92-96: The otolith works comes in like an afterthought here, since it is not set up in any way in the Introduction section (linked to major comment regarding otolith work)

Done, we have now changed this part about the otolith analysis in the whole paper.

- Section 2.1: more clearly point out that this (or very similar) cod condition time series were previously published and are here updated to 2018?

Done.

- LL107-109: please explain rationale of using size class 20-29 and 40-49 cm for condition calculations.

Done.

- Section2.2: suggest to point out more clearly the key difference between studies, focus on fall here versus all seasons in Orio et al 2019 (see my previous comment above).

Done.

- LL125-135: I am not a physiologist, but I guess in principle use of oxygen as continuous variable (instead of somewhat arbitrary boundaries) would make sense. I can see that use of specific limits facilitates analysis, but would mention this possible limitation.

The sub-lethal boundary we used (4 ml/l) is from the experiment by Chabot and Dutil (1999), it is not arbitrary. However, we have now also shown the actual oxygen experienced by the population. About the boundary 1 ml/l (avoidance), it is a well known boundary for Baltic Sea cod (Schaber et al. 2012).

- L155: Explain the rationale of using a Fulton's k of 0.9. Also give other thresholds(e.g., "very low" used later in L163) here already.

This part has now been deleted and we refer now to the results of Limburg and Casini

(2019).

- Section 3.2: in the Discussion section (not here), suggest to discuss the patterns observed for fall here compared to the patterns in Orio et al 2019 reporting cod depth distribution contraction to shallower water for SD26-28 when looking at the entire year.

Done.

- Section 3.2, 3.3, 3.4: would all benefit a lot from formal statistics.

We agree that the trends of the depth patterns are not linear, that is also why a statistical tests of the temporal patterns would not provide much additional information in our opinion. We now estimated the actual oxygen that the population has been experiencing over time (not only the overlap with low-oxygen levels below a certain threshold) and we perform statistical analysis relating this with fish condition.

- L243: I think the discussion of mechanisms that can explain what drives cod into layers with low oxygen levels is quite central, since it relates to the key novel finding of this manuscript. Suggest to therefore not state that "beyond scope" of manuscript, but rather state that you can only speculate and will discuss possible causes as systematically as possible.

Our paper is focusing on showing the processes explaining the link between the general Baltic deoxygenation and condition (as shown by Casini et al. (2016)) and putting in a population context what found previously in the cod otoliths by Limburg & Casini (2019). Therefore to link the population overlap with low-oxygen waters with fish condition. We really think that explaining the reasons why cod move deeper in autumn deserve a full analysis and this is beyond our scope. We have however provided a potential explanation to the deepening of the distribution in the paper and we say that focused analyses should be done to provide an answer to this interesting question.

- LL244-245: The role of temperature was also the first thing that came to my mind, but I then wondered about actual temperature profiles in fall, and whether they would

support these considerations. It would be useful to include information on prevailing temperature depth profiles in fall as background for the discussion.

We have now added temperature and salinity data to support the Discussion.

- L283: Should read "although we have confirmed here that ..." and refer to Limburg and Casini 2019.

In Limburg & Casini (2019) it was shown that fish in low condition at capture were exposed during their lives to lower oxygen levels than those in good condition (at least from the mid-1990s), without saying anything about the distribution of the population, and therefore whether or not a large part of the population indeed experienced stressful circumstances.

- LL283-291: Discussion of other factors could be more extensive. Cite Brander et al 2020 here as well.

Done.

- L297: Agree, very interesting future direction, and a question that really results for the first time from the analyses in this manuscript (not possible from Limburg and Casini 2019) – this would be worth pointing out.

We have now deleted the otolith part and we think the sentence can keep as it is.

- Figures: I suggest to add a figure to illustrate key findings regarding the correlation of cod condition and the overlap of cod depth distribution and low oxygen waters.

We have now analysed the relation between the actual oxygen experienced and condition that will change somewhat the disposition of the figures.

- Related to the general comment regarding the presentation of otolith microchemistry data in this manuscript, Figure 6 of this manuscript appear to be an alternative view of Figure 2 c in Limburg and Casini 2019, i.e., not adding new information here that could not be provided from that manuscript.

[Figure]

We have now deleted the otolith analysis and referred to Limburg and Casini (2019) instead.

Technical corrections

- LL23-24, L62: wording should be more precise – "exponential increase" not really correct, suggest "strong increase"; "largest marine dead zone", unnecessarily dramatic.

We agree about the first suggestion, but not about the second since the low-oxygen zones are called indeed "dead zones" in literature.

- LL26: "elusive" does not really reflect that specific alternative mechanisms have been proposed.

Here we meant, as stated, that the processes behind the statistical relation between general hypoxia and cod population condition found previously remained elusive.

- LL29-32: rephrase, confusing wording.

Done.

- L59-60: Wording in Breitburg et al 2018 is more scientific ("low O2 areas have become more extensive and severe") – suggest to follow this approach.

Dead-zones is a term used commonly in literature, named also in Breitburg et al. (2018), we prefer to keep this terminology.

- L81: start new paragraph, focusing on effects and not mechanisms from here on.

Done.

- L82-82: rephrase "lamented"

Done.

- L194: "in a couple..." – word missing?

Correct, done.

- L254: "hostile waters" – suggest to rephrase

We would like to keep this wording, giving a clear idea of the concept.

References

Breitburg, D., Levin, L. A., Oschlies, A., Grégoire, M., Chavez, F. P., Conley, D. J., Garçon, V., Gilbert, D., Gutiérrez, D., Isensee, K., Jacinto, G. S., Limburg, K. E., Montes, I., Naqvi, S. W. A., Pitcher, G. C., Rabalais, N. N., Roman, M. R., Rose, K. A., Seibel, B. A., Telszewski, M., Yasuhara, M., and Zhang, J. 2018. Declining oxygen in the global ocean and coastal waters. Science, 359, eaam7240. Doi:10.1126/science.aam7240.

Casini, M., Käll, F., Hansson, M., Plikshs, M., Baranova, T., Karlsson, O., Lundström, K., Neuenfeldt, S., Gårdmark, G., and Hjelm J. 2016a. Hypoxic areas, density dependence and food limitation drive the body condition of a heavily exploited marine fish predator. R. Soc. Open Sci., 3, 160416.

Chabot, D., and Dutil, J.-D. 1999. Reduced growth of Atlantic cod in non-lethal hypoxic conditions. J. Fish Biol., 55, 472–491.

Limburg, K., and Casini, M. 2019. Otolith chemistry indicates recent worsened Baltic cod condition is linked to hypoxia exposure. Biol. Lett., 15, 20190352.

Schaber, M., Hinrichsen, H.-H, and Gröger, J. 2012. Seasonal changes in vertical distribution patterns of cod (Gadus morhua) in the Bornholm Basin, Central Baltic Sea. Fish. Oceanogr. 21, 33–43.
* * *

---

## Author Comment (AC6) · 11 Jun 2020

We believe this comment has the same content as the first comment (RC1) from the same Reviewer. Our reply was provided to RC1.
* * *

---

## Author Response (AR1)

We have now addressed the comments and suggestions of the three reviewers. The major changes are:
1) we have explained in more details some of the data, 2) we have added to the paper the oxygen
actually experienced by cod during the years; 3) we have deleted the part on otolith analyses and
related our findings to those by Limburg and Casini (2019) in the Discussion, 4) we have performed a
statistical GAM analyses of the relation between cod condition and oxygen, and 5) we have further
improved the Discussion with more discussion about the differences in depth distribution between our
study and Orio et al. (2019), the findings of Brander (2020) and the other potential reasons behind cod
condition decline, as requested by the reviewers.

**Reply to reviewer #1**

*We thank the reviewer for his thorough comments.*

*In literature, there have been only two studies investigating the relation between Baltic deoxygenation
and cod condition, i.e. Casini et al. (2016) and Limburg & Casini (2019). In the former paper, a
strong correlation was found between the extent of hypoxic areas (defined in that paper as $km^2$ with
oxygen < 2 ml/l) and condition, but the mechanisms potentially explaining the statistical relationships
were not investigated but just proposed, i.e. decline in benthic food, changes in cod
behavior/distribution, direct physiological stress, or of course a combination of these. In the second
paper (Limburg & Casini 2019) it was shown that fish in low condition at capture were exposed
during their lives to lower oxygen levels than those in good condition (at least from the mid-1990s),
without saying anything about the distribution of the population, and therefore whether or not a large
part of the population indeed experienced stressful circumstances. Therefore, the original triggers
and the mechanisms relating hypoxia to the average Baltic cod condition in the population were
indeed elusive (and we think they still need attention), as we state in the abstract of the new paper
(referred to as CHOL, from the initial of the authors names, following the terminology of the
Reviewer; we refer here to the Reviewer as KB).*

*The CHOL paper takes a further step, showing that the cod population went progressively deeper in
autumn and this, concomitant with a shallowing of the low-oxygen layers, increased the spatial
overlap between cod distribution and low-oxygen waters, and thus generating stressful circumstances
for the cod population (exposure to waters with oxygen < 4 ml/l, detrimental for cod condition as
found in experiments by Chabot & Dutil, 1999) (see below about the choice of the oxygen sub-lethal
threshold in the CHOL paper). We finally showed that this increased overlap relates statistically to
the decline in the mean population condition and to the proportion of fish with very low condition,
both for juveniles and large fish. Therefore, the CHOL paper shows the original processes (deepening
of the cod population concurrent with the shallowing of low-oxygen layers) creating the stressful
circumstances relating to a decline in condition, for both small and large cod. In our opinion, this is a
very important step forward in the understanding of the link between low-oxygen and cod condition,
and in general for understanding the causes of the declined cod condition. Additionally, it is not so
obvious that condition has to be directly linked to a general deoxygenation phenomenon, since mobile
fish can change their distribution in response to that, as done by other fish species in other areas.
This did not happen for the Baltic cod (conversely it went deeper, in autumn), and we think that
finding the answer to why this has happened is one of the next challenges for the scientists. Cod prey
should also suffer from deoxygenation, although some are more tolerant to low oxygen; therefore, the
question of why cod went deeper is not so trivial in our opinion and should be investigated as we
suggested for future studies.*

*We are therefore totally in line with KB about the fact that "the issues to resolve are firstly whether
cod redistribute themselves to remain in areas and depths with sufficient oxygen and if not then*

*secondly whether the magnitude of ambient oxygen decline that cod experience is sufficient to explain*
*all or only part of the observed change in their condition." This is exactly what we have done in the*
*paper for both small and large cod in autumn. In addition, we have also investigated the original*
*reasons creating these circumstances (i.e. both deepening of the population and shallowing of the*
*low-oxygen layers), as well as estimated the overlap with the low-oxygen layers, known to affect cod*
*condition, and estimated the relation between this overlap, the mean population condition and the*
*percentage of fish with very low condition, for both small and large cod. This does not mean that*
*direct exposure to low-oxygen is the sole driver of condition (even if oxygen decline is sufficient to*
*explain a large part of the decline in condition), because there can be other contributing drivers*
*and/or drivers that have co-varied with deoxygenation (food availability, parasites, inter- and intra-*
*specific competition, etc…) that could also explain the reduced feeding level (see below). That is why*
*further work is needed here too.*

*In the CHOL paper, we used 4 ml/l as sub-lethal oxygen threshold impairing cod condition. As KB*
*correctly stated, 73% oxygen saturation (sub-lethal threshold in Chabot & Dutil (1999)) corresponds*
*to 4.8 ml/l at the experimental conditions, but 65% oxygen saturation is the level from which the*
*decline in condition was significant in Chabot & Dutil (1999) experiment, corresponding to 4.3 ml/l.*
*We therefore used now 4.3 in the revised version of the paper, to improve our analyses.*

*We agree with KB that the real oxygen levels experienced by cod would be informative, so in the*
*revised paper we showed also the oxygen levels corresponding to the annual depth distribution of cod*
*in autumn, both for small and large cod. However, we think that the information about the shallowing*
*of the 1 ml/l and 4 ml/l (the latter now 4.3 ml/l in the revised paper) depths enriches the story, and*
*together with the deepening of the cod distribution depth, it visually delivers a very clear message.*
*Therefore, we preferred to retain it.*

*The otolith analysis was already published in Limburg and Casini (2019), but the analysis was re-*
*arranged as a new figure in CHOL. We thought that this was a nice conclusion of the story, but we*
*have now opted to delete it from the paper and to discuss instead the results in Limburg and Casini*
*(2019) in our paper.*

*Exploring mechanistic relationships would need experimental setups. Using time-series, the statistical*
*relationships have to be interpreted in light of what is known about the biology and ecology of the*
*fish. In our case, we used the experimental results from Chabot & Dutil (1999) to relate the*
*distribution of the cod population with the oxygen levels resulted to affect cod in experimental setups,*
*and in the revised paper we have briefly discussed the information coming from stomach content*
*analysis (see below).*

*We agree with KB that Neuenfeldt et al. (2020) is an extremely important paper, showing that the*
*lower energy intake observed in cod (using stomach content time-series) would predict a decrease*
*growth in length that could explain the shift in size distribution of cod population towards lower sizes.*
*The lower amount of benthos and pelagic fish in the diet of cod could be due to a decline in their*
*availability (as suggested in Neuenfeldt et al. 2020) but also to a decline in cod appetite due to low-*
*oxygen exposure (Chabot & Dutil 1999, Brander 2020) or other low oxygen-related physiological*
*stress. Food intake can surely be the main driver of growth, but other factors can cause fish to*
*allocate more energy to basic metabolism, reproduction etc… in some circumstances. For example,*
*currently Baltic cod reproduce at a smaller size (around 20 cm) than before (30-35 cm) and this could*
*mean a lower allocation of energy to growth and therefore also explain the growth decline. We agree*
*with KB that such reasoning produces the egg-chicken problem, but it brings us outside the scope of*
*the CHOL paper.*

*In our analyses we investigate fish condition, not growth in length, and since the two traits are*
*different (fish can grow fast in length, utilizing the stored energy reserves, but this at the detriment of*

*condition, that is a ratio between weight and length, as shown also in feeding experiments) we do not*
*want to mix them. However, in the revised paper, we have added more discussion about the decline in*
*feeding level found in Neuenfeldt et al. (2020) that could link the increased exposure to low-oxygen*
*levels to declined condition. However, there are some aspects that make this link not as straight*
*forward as it seems. Neuenfeldt et al. (2020) show that feeding level has not declined for large cod,*
*but the observed decline in condition has been more severe for large cod (Casini et al. 2016 and the*
*new CHOL paper), suggesting perhaps that feeding level is not the sole driver of large cod condition*
*and that therefore low oxygen has impacted cod condition also through different mechanisms, other*
*than food intake. For example, large cod could experience shortage of benthic prey and therefore,*
*proportionally, could be forced to eat more pelagic fish that require higher energy to catch.*
*Moreover, cod was not in low-oxygen conditions before the early 1990s (see our CHOL paper), but*
*the feeding level was already low (Neuenfeldt et al. 2020; see also ICES 2016), and so was condition*
*(Casini et al. 2016, new CHOL paper), indicating that direct exposure to hypoxia is not always the*
*driver of feeding level and condition (matching therefore with the results from otolith analyses in*
*Limburg & Casini (2019)). In the revised paper, we have however put our results in relation to*
*Neuenfeldt et al. (2020) findings about feeding levels, to link the increased overlap with low-oxygen*
*waters to feeding level and condition after the early 1990s. We have moreover related more the*
*CHOL paper results with Brander (2020) paper recently published.*

- Minor editorial comments line 23 and 62 -The expansion of hypoxic areas has been quite rapid, but
not exponential line 76 – make it clear that this explanation is inference and not based on evidence
line 178 "these" presumably refers to "large fish" - better to say so.

*We have now edited these specific points.*

*Reply to reviewer #2*

*We thank the reviewer for the helpful comments.*

 *Reply to the general comments:*

- The paper is interesting, and the patterns are convincing. Inevitable any conclusions drawn from
parallel changes in two or more metrics without a test will be speculative. Nevertheless, I think the
authors do a good enough job of highlighting hypoxia as a contributor to decreasing cod condition.
However, I think the description on confounding effects and other contribution factors could be
improved. For example, although hypoxia may well contribute greatly to low growth of cod in the
current system the drivers of a decrease in condition are the triggers of a change in depth distribution
and the cause of low oxygen.

*We present briefly the potential other factors contributing to the cod condition patterns also in the*
*Introduction to provide some background, specifying that in literature deoxygenation has been*
*advocated as one of the major drivers of the condition decline (e.g. Casini et al. 2016). Our present*
*manuscript is focusing on showing the processes (deepening of cod population and shallowing of low-*
*oxygen layers) explaining the link between the general Baltic deoxygenation and condition (as shown*
*by Casini et al. (2016)) and putting in a population context what found previously in the cod otoliths by*
*Limburg & Casini (2019). We appreciate the reviewer's point, and thus we added some text in the*
*Discussion section about the alternative factors that could contribute to explain the patterns in cod*
*condition. On the other hand, the reasons for the deepening of the cod population have not been*
*investigated, only speculated about in other papers (Orio et al. 2019). These are beyond the scope of*
*our paper, but we have suggested that this is an important question to answer in futures studies.*

- Furthermore, there is no description of any statistical analysis. Mostly the patterns are "analyzed" by
eye and described in the results chapter (related note referring to figures as you describe results). This
approach may occasionally be valid – and the patterns described are convincing enough - but at least
some sort of quantification of the size of effects across time should use when describing them (reduced
from x to x). A statistic test is used for the otolith data, but this is not included in the methods. The
results from the otolith analysis is interesting yet this part of the paper is referred to as an afterthought
throughout the paper. I think this analysis warrants increased value, both by adding to the introduction
enough background material to allow readers to evaluate the validity of the methods on know of any
prior findings and in the a fuller description of methods including how the otoliths were selected.

*In the revised manuscript we have been more quantitative, spelling out the most important changes*
*across time. We now also estimated the actual oxygen that the population has been experiencing over*
*time (not only the overlap with low-oxygen levels below a certain threshold) and we perform statistical*
*analysis relating this with fish condition. The otolith analysis has been now deleted from the paper, as*
*suggested by the other reviewers, and we now discuss our results in view of what found by Limburg and*
*Casini 2019.*

*Reply to the specific comments:*

- 26+28: Is "processes" the right word?

*We have changed the wording now.*

- 100: What is the sample size?

*We have added it.*

- 101: is this data stable once entered, or is it subject to change? In the last case, a date of retrieval would
be handy to include.

*We have now added the date of data extraction for the years after 1990, which can undergo slight*
*updates in the ICES DATRAS database. The years before 1990s are from historical databases and*
*therefore not subject to changes.*

- 105: there are different ways to measure 'total length', maybe explain in more detail how it was done
in this study.

*Done.*

- 107: why is SD26-28 chosen and not for example not 29?

*We have now explained the reason.*

- 108/109: why is the subdivision of big and small cod made and why those specific lengths? What
happens with fish between 29 and 40 cm?

*The two length groups for condition were selected to represent small and large fish, as stated in the*
*paper. The small fish can also be seen as juveniles even though the size at maturity has declined with*
*time for this population. The large fish on the other hand can all be considered adults. Currently, there*
*are very few cod above 50 cm and therefore we could not use larger size-classes. We have now been*
*more specific in this part.*

- 109: Quarter 4 also includes part of the winter. Why not mentioning the exact months instead of season
or quarter 4?

*Done*

- 117: why are those class divisions different from row 108?

*The population distributions, divided in < 30cm and ≥ 30 cm, come from Orio et al. (2019). In the*
*condition estimations, we did not want to use too large ranges of fish sizes in one group because Fulton*
*condition factor (used in the paper we refer to and compare to ours) can be affected by fish size.*
*Moreover, cod start to become piscivorous around 30 cm and therefore fish below 30 cm (but larger*
*than in the plankton- and nektobenthos-feeder phase, around 15 cm) can be considered occupying*
*similar ecological niche. Therefore, the 20-29 and 40-49 size groups were chosen for condition*
*estimation just to represent the small and large sizes with different ecological niches and therefore*
*likely different behavior and food requirements. We have now added some clarification into this part.*

- 135: it is later explained, but I would rather put here the <0.8 (Eero et al 2012), explaining the 'very
low' condition

*Done*

- 160-191: I see many statements as 'more' and 'lower' and 'deeper', but it is very descriptive, and I
miss actual numbers in some places and statistical tests to prove these statements. Also, how many data
points were retrieved, how big was the sample size?

*We do not think we need statistical tests to explain the long-term patterns, what is important is the*
*overlap between the cod population and low-oxygen layers. However, we tried to add some more*
*quantitative information in the text. We now also estimated the actual oxygen that the population has*
*been experiencing over time (not only the overlap with low-oxygen levels below a certain threshold)*
*and we perform statistical analysis relating this with fish condition. We have also added the samples*
*sizes in the Methods.*

- 171: which depth?

*Done, we have improved this description.*

- 186: The oxygen layers are almost the same, but not totally. I understand this is because they are
weighed with the SD-specific distribution of the cod, but I think it makes things clearer if you write somewhere that this means that it differs between the big and the small cod (it took me a while to understand).

*Done.*

- 267: I miss a note about that it is not 100% sure that the cod are actually in those low oxygen waters, because that was not directly measured. However, the additional otolith results make it very plausible that this is the case.

*We agree, fish can move and therefore we cannot be sure that those with very low condition spent most of their time in low-oxygen waters (even if they were caught there) from the time-series, but as the Reviewer #2 also says, this is very plausible also considering the results of the otoliths' analyses in Limburg and Casini (2019).*

- 273: Was there a way to directly link otolith chemistry with body condition? (e.g. from the same individual?) Why do you think the overlap between cod and oxygen layers is oscillating? (why is the oxygen stratification oscillating?)

*Yes, it is possible analyzing the Mn/Mg elements ratio in the otoliths of individual fish, see Limburg & Casini (2018, 2019).*

- 475/476/481/486: you use here the whole word 'subdivision', while in the previous description (472) you al-ready used SD

*We have edited this to be more consistent.*

- 490 post-2000? This is differently described throughout the text.

*We have now deleted the part about otolith analyses, referring instead to Limburg and Casini 2019, as suggested by the other reviewers.*

- Figure 3: Is there a possible explanation for the high condition in 1996 in SD25

*In general, the mid 1990s are characterized by good oxygen conditions (low extent of hypoxic areas) and a large increase in the sprat stock, probably boosting condition. We feel that going into these details bring us out of the paper's scope and we prefer not to focus on single annual values but on the general patterns.*

- Figure 6: 2000 onward is called 'post 2000' in the text. Why are there squares in the boxes

*We have now deleted the part about otolith analyses, referring instead to Limburg and Casini 2019, as suggested by the other reviewers.*

***Reply to reviewer #3***

*We thank the reviewer for the helpful comments.*

*Reply to the general comments:*

- Tendency to oversell the results: this is recurring through the title, abstract and discussion. For
example, the study title is not in line with the results. The title implies that the study results alone explain
the low condition of Baltic cod, when in reality, the study sheds additional light on one potential
mechanism, direct exposure to low oxygen waters, which does not rule out alternative mechanisms
(both linked to expanding oxygen minimum zones and to other factors) that have been proposed before.
Abstract L27-29: point out more clearly that the study is assessing the role of direct exposure to low
oxygen waters, not "the processes". Discussion L223: should be "one mechanism" not "the
mechanisms". Conclusion L293: should be "shown here one mechanism", not "the mechanisms".

*In the abstract and discussion, when we speak about processes and mechanisms we refer to the*
*shallowing of low-oxygen layers and deepening of the cod population that create the overlap and*
*therefore the circumstances for a direct exposure effect. However, we have now gone through the*
*manuscript and edited some sentences not to oversell our results.*

- Delineation of results from previous work: Exposure to low oxygen water was already previously
linked to the Baltic cod condition decline by Limburg and Casini (2019) using otolith microchemistry.
This study is cited and referred to by Casini et al., but still, the apparent narrative here is that the
exposure to low oxygen waters is shown via the identification of increasing overlap of the depths of low
oxygen waters and the cod depth distribution, and that this is then confirmed with otolith
microchemistry in this manuscript (e.g., abstract LL 29-34, Introduction LL 89-93, Discussion LL 224-
228). This really has it backwards. I suggest to instead clearly lay out key results and conclusions from
Limburg and Casini 2019 in the Introduction, and then use this as rationale for the (relevant and
interesting) independent confirmation and new insights into the specific patterns of exposure to low
oxygen waters in this manuscript.

*We have now followed the suggestion from the Reviewer, specifying in the Introduction that in Limburg*
*& Casini (2019) it was shown that fish in low condition at capture were exposed during their lives to*
*lower oxygen levels than those in good condition (at least from the mid-1990s), without saying anything*
*about the distribution of the population, and therefore whether or not a large part of the population*
*indeed experienced stressful circumstances, that could explain the low population condition found in*
*Casini et al. (2016).*

- Use and presentation of otolith microchemistry dataset from Limburg and Casini 2019 in this study
(connected to previous comment): I would strongly recommend the exclusion of these data from the
present manuscript. To me, the analysis and results mirror the previous publication by Limburg and
Casini too closely to warrant inclusion here. The authors acknowledge the previous study, but without
going into details. However, the dataset, analyses, discussion points (Section4.2) and conclusions are
largely the same. Also, the results from otolith microchemistry analyses are not formally correlated to
the depth distribution analyses, and appear rather like an "afterthought" in this manuscript. The
inclusion in the manuscript thus unnecessarily duplicates previous work. If conclusions from the
previous work are instead clearly presented in the Introduction, this will provide the rationale for the
real strength and novelty of the present study, the depth distribution analyses. New insights from this independent approach compared to the insights from the original otolith microchemistry approach could
then also be discussed more explicitly in the Discussion. Interestingly, all conclusions in the conclusion
section of the manuscript (LL293-306) relate to this aspect of the study anyway.

*We have now followed the suggestion from the Reviewer, specifying in the Introduction that in Limburg*
*& Casini (2019) it was shown that fish in low condition at capture were exposed during their lives to*
*lower oxygen levels than those in good condition (at least from the mid-1990s), without saying anything*
*about the distribution of the population, and therefore whether or not a large part of the population*
*indeed experienced stressful circumstances, that could explain the low population condition found in*
*Casini et al. (2016).*

- Statistical analyses: Right now, the manuscript is lacking in formal statistical assessments. This
includes statistical approaches to assess the significance and nature of temporal trends in the depths of
low oxygen waters, cod depth distributions and overlap, as well as the formal assessment of the link of
overlap and cod condition over time. The Material and Methods should then also include a dedicated
section outlining statistical approaches. In this context, looking at Figure 4 of the manuscript, many of
the observed temporal changes do not look linear. E.g., for SD26-28, cod mean depth was essentially
stable after 1990, and for SD25, neither cod depth distribution nor depth of low oxygen water appears
to change significantly between 2008 and 2018. Formal statistical analysis would therefore have the
potential to lead to additional insights beyond the points included in the manuscript.

*We agree that the trends of the depth patterns are not linear, that is also why standard statistical tests*
*of the temporal patterns would not provide much additional information in our opinion. We now*
*estimated the actual oxygen that the population has been experiencing over time (not only the overlap*
*with low-oxygen levels below a certain threshold) and we perform statistical analysis relating these 2*
*metrics (overlap and actual oxygen concentration) with fish condition.*

*Reply to the specific comments:*

- Throughout the entire manuscript, I was waiting for an explanation for the discrepancy of the cod
depth distribution trends over time between the very similar data sets and analyses in Orio et al. 2019
(showing cod distributions at least for SD26-28 becoming shallower since the 1990s) and this
manuscript. This was then given in the second to last sentence of the conclusions:) I suggest to explicitly
explain the difference between the datasets (fall versus other seasons) already in the Material and
Methods, and then discuss this interesting difference between seasons in the main part of the Discussion,
not just in the Conclusion.

*We present now a larger discussion of these seasonal differences in the Discussion. We do not think*
*explaining this in the Material and Methods is necessary, since we are very clear that our paper is*
*focusing on Quarter 4.*

- L53: Would cite Chabot and Dutil 1999 here already.

*Done*

- L60: Suggest addition of Reusch et al 2018 as probably best reference for combined strong temporal
changes in temperature, eutrophication, oxygen in the Baltic Sea.

*Done*

- L60-61: to my knowledge, the degradation of benthic communities is NOT well documented in the
Baltic Sea, and lack of time series on benthic communities has been one of the issues hampering
understanding of consequences of expanding oxygen minimum ones. Rephrase.

*Done*

- L71: see major comments regarding previous results from Limburg and Casini 2019. Suggest to
present in much more depth here and explain that link between low condition and exposure to low
oxygen water was established in that study.

*Done. We have now specified in the Introduction that in Limburg & Casini (2019) it was shown that*
*fish in low condition at capture were exposed during their lives to lower oxygen levels than those in*
*good condition (at least from the mid-1990s), without saying anything about the distribution of the*
*population, and therefore whether or not a large part of the population indeed experienced stressful*
*circumstances, that could explain the low population condition found in Casini et al. (2016).*

- L73: suggest to mention the actual mechanism connected to this, density dependence.

*Done, but we also meant change in the habitat occupation, not only contraction, we have now rephrased*
*the sentence.*

- L73-75: add mechanism proposed by Brandner 2020, mild hypoxia reducing rate of digestion.

*We consider this mechanism already included in the sentence (stress due to hypoxia exposure). This*
*part of the Introduction has been however changed extensively now.*

- L92-96: The otolith works comes in like an afterthought here, since it is not set up in any way in the
Introduction section (linked to major comment regarding otolith work)

*We have now removed the part about the otolith analysis from the paper, and instead discussed our*
*results in view of Limburg & Casini (2019) in the Discussion.*

- Section 2.1: more clearly point out that this (or very similar) cod condition time series were previously
published and are here updated to 2018?

*Done.*

- LL107-109: please explain rationale of using size class 20-29 and 40-49 cm for condition calculations.

*Done.*

- Section2.2: suggest to point out more clearly the key difference between studies, focus on fall here
versus all seasons in Orio et al 2019 (see my previous comment above).

*We already stated here that we used the model depth estimates in Quarter 4, consistent with the oxygen*
*used in the study. We prefer to speak about the differences with Orio et al. 2019 in the Discussion.*

- LL125-135: I am not a physiologist, but I guess in principle use of oxygen as continuous variable
(instead of somewhat arbitrary boundaries) would make sense. I can see that use of specific limits
facilitates analysis, but would mention this possible limitation.

*The sub-lethal boundary we used (4 ml/l, now more precisely set at 4.3 ml/l following Reviewer #1*
*comments) is from the experiment by Chabot and Dutil (1999), it is not arbitrary although based on*
*cod from another region. About the boundary 1 ml/l (avoidance), it is a well known boundary for Baltic*
*Sea cod (Schaber et al. 2012). However, we have now also shown, and made analyses with, the actual*
*oxygen experienced by the population.*

- L155: Explain the rationale of using a Fulton's k of 0.9. Also give other thresholds(e.g., "very low"
used later in L163) here already.

*This part has now been deleted and we refer now to the results of Limburg and Casini (2019).*

- Section 3.2: in the Discussion section (not here), suggest to discuss the patterns observed for fall here
compared to the patterns in Orio et al 2019 reporting cod depth distribution contraction to shallower
water for SD26-28 when looking at the entire year.

*Done.*

- Section 3.2, 3.3, 3.4: would all benefit a lot from formal statistics.

*We agree that the trends of the depth patterns are not linear, that is also why standard statistical tests*
*of the temporal patterns would not provide much additional information in our opinion. We now*
*estimated the actual oxygen that the population has been experiencing over time (not only the overlap*
*with low-oxygen levels below a certain threshold) and we performed statistical analysis relating these*
*metrics (overlap and actual oxygen experienced) with fish condition.*

- L243: I think the discussion of mechanisms that can explain what drives cod into layers with low
oxygen levels is quite central, since it relates to the key novel finding of this manuscript. Suggest to
therefore not state that "beyond scope" of manuscript, but rather state that you can only speculate and
will discuss possible causes as systematically as possible.

*Our paper is focusing on showing the processes explaining the link between the general Baltic*
*deoxygenation and condition (as shown by Casini et al. (2016)) and putting in a population context*
*what found previously in the cod otoliths by Limburg & Casini (2019). Therefore to link the population*
*overlap with low-oxygen waters with fish condition. We really think that explaining the reasons why*
*cod move deeper in autumn deserve a full analysis and this is beyond our scope. We have however*
*provided a potential explanation to the deepening of the distribution in the paper and we say that*
*focused analyses should be done to provide an answer to this interesting question.*

- LL244-245: The role of temperature was also the first thing that came to my mind, but I then wondered
about actual temperature profiles in fall, and whether they would support these considerations. It would
be useful to include information on prevailing temperature depth profiles in fall as background for the
discussion.

*Since we do not deal with the reasons of the increased depth of the cod population, we prefer not to*
*present temperature (or other information), that would be incomplete to make such analyses on draw*
*conclusions. We think that a focused analysis should be done to answer this question.*

- L283: Should read "although we have confirmed here that ..." and refer to Limburg and Casini 2019.

*Done.*

- LL283-291: Discussion of other factors could be more extensive. Cite Brander et al 2020 here as well.

*Done.*

- L297: Agree, very interesting future direction, and a question that really results for the first time from
the analyses in this manuscript (not possible from Limburg and Casini 2019) – this would be worth
pointing out.

*We have now deleted the otolith part. We think that the novelty of this new paper (the finding that the*
*population has been progressively more experiencing low-oxygen waters, and that this was due to both*
*a shallowing of low-oxygen layers and deepening of the population) is now clear.*

- Figures: I suggest to add a figure to illustrate key findings regarding the correlation of cod condition
and the overlap of cod depth distribution and low oxygen waters.

*We have now analysed the relation between the actual oxygen experienced and condition that will*
*change somewhat the disposition of the figures.*

- Related to the general comment regarding the presentation of otolith microchemistry data in this
manuscript, Figure 6 of this manuscript appear to be an alternative view of Figure 2 c in Limburg and
Casini 2019, i.e., not adding new information here that could not be provided from that manuscript.

*We have now deleted the otolith analysis and referred to, and discussed the results of, Limburg and*
*Casini (2019) instead.*

Technical corrections

- LL23-24, L62: wording should be more precise – "exponential increase" not really correct, suggest
"strong increase"; "largest marine dead zone", unnecessarily dramatic.

*We agree about the first suggestion, but not about the second since the low-oxygen zones are called*
*indeed "dead zones" in literature.*

- LL26: "elusive" does not really reflect that specific alternative mechanisms have been proposed.

*Here we meant, as stated, that the processes behind the statistical relation between general hypoxia*
*and cod population condition found previously remained elusive.*

- LL29-32: rephrase, confusing wording.

*Done.*

- L59-60: Wording in Breitburg et al 2018 is more scientific ("low O2 areas have become more
extensive and severe") – suggest to follow this approach.

*Dead-zones is a term used commonly in literature, named also in Breitburg et al. (2018), we prefer to*
*keep this terminology here.*

- L81: start new paragraph, focusing on effects and not mechanisms from here on.

*Done.*

- L82-82: rephrase "lamented"

*Done.*

- L194: "in a couple..." – word missing?

*Correct, done.*

- L254: "hostile waters" – suggest to rephrase

*We would like to keep this wording, giving a clear idea of the concept.*

*References*

[revised manuscript text omitted]

Figure 1

[Figure]

Figure 2

[Figure]

Figure 3

[Figure]

Figure 4

[Figure]

Figure 5

[Figure]

Figure 6

**A**

[Figure]

**B**

[Figure]

---

## Author Response (AR2)

We have now addressed the comments and suggestions of the Editor and the two reviewers. Specifically, we have 1) edited the title of the paper and 2) adjusted some grammar and typos along the text. We have also 3) modified the title of the main y-axis in Figs. 4 and 5 (and the respective captions), now matching with the terminology in Fig. 6, to better reflect their content and to avoid misinterpretations.